# CREBBP inactivation sensitizes B cell acute lymphoblastic leukemia to ferroptotic cell death upon BCL2 inhibition

Alicia Garcia-Gimenez[1,2], Jonathan E. Ditcham [1,2], Dhoyazan M. A. Azazi [1,2], George Giotopoulos [1,2], Ryan Asby[1,2], Eshwar Meduri[1,2], Jaana Bagri[1,2], Nathalie Sakakini[1,2], Cecile K. Lopez[1,2], Nisha Narayan [1,2], Tumas Beinortas[1,2,3], Shuchi Agrawal-Singh[1,2,4], Kent Fung[5], David O'Connor[5], Marc R. Mansour [5,6], Husam B. R. Alabed[7,8], Benjamin Jenkins [7], Albert Koulman [7], Michael P. Murphy [9], Sarah J. Horton [1,2], Brian J. P. Huntly [1,2,3,10] ✉ & Simon E. Richardson [1,2,3,10] ✉

B-cell acute lymphoblastic leukemia (B-ALL) is a leading cause of death in childhood and outcomes in adults remain dismal. There is therefore an urgent clinical need for therapies that target the highest risk cases. Mutations in the histone acetyltransferase *CREBBP* confer high-risk and increased chemoresistance in ALL. Performing a targeted drug-screen in isogenic human cell lines, we identify a number of small molecules that specifically target *CREBBP*-mutated B-ALL, the most potent being the BCL2-inhibitor Venetoclax. Of note, this acts through a non-canonical mechanism resulting in ferroptotic rather than apoptotic cell death. *CREBBP*-mutated cell lines show differences in cell-cycle, metabolism, lipid composition and response to oxidative stress, predisposing them to ferroptosis, which are further dysregulated upon acquisition of Venetoclax resistance. Lastly, small-molecule inhibition of CREBBP pharmacocopies *CREBBP*-mutation, sensitizing B-ALL cells, regardless of genotype, to Venetoclax-induced ferroptosis in-vitro and in-vivo, providing a promising drug combination for broader clinical translation in B-ALL.

B-cell acute lymphoblastic leukemia (B-ALL) is an aggressive hematological malignancy of B-lineage progenitors and is the commonest cancer in children[1]. Whilst the majority of children can be cured with multi-agent chemotherapy, patients with high-risk genetic subtypes, certain age groups and those who relapse remain a clinical challenge, such that B-ALL remains a leading cause of death in childhood. Furthermore, outcomes of adults with B-ALL remain dismal, even when fit enough to be treated intensively. There is therefore an urgent need to better understand drivers of high-risk B-ALL and to develop novel therapeutic approaches targeting these challenging patient cohorts.

Mutations in *CREBBP* (CREB-binding protein) are found in multiple hematological and solid malignancies, notably B-cell lymphomas[2,3]. Loss-of-function (LOF) mutations affecting *CREBBP* are also recurrent second-hit mutations across multiple genetic subtypes of B-ALL and

[1]Department of Haematology, Cambridge Stem Cell Institute, Cambridge, UK. [2]Cambridge Stem Cell Institute, Cambridge, UK. [3]Cambridge University Hospitals, Cambridge, UK. [4]Centre for Haematology, Department of Immunology and Inflammation, Imperial College London, London, UK. [5]University College London Cancer Institute, UCL, London, UK. [6]UCL Great Ormond Street Institute of Child Health, London, UK. [7]Institute of Metabolic Science, University of Cambridge, Cambridge, UK. [8]Department of Chemistry, Biology and Biotechnology, University of Perugia, 06100 Perugia, Italy. [9]MRC Mitochondrial Biology Unit, Keith Peters Building, University of Cambridge, Cambridge, UK. [10]These authors jointly supervised this work: Brian J. P. Huntly, Simon E. Richardson. ✉e-mail: bjph2@cam.ac.uk; ser32@cam.ac.uk

are associated with adverse features, including high-risk genetic subtypes and persistent measurable residual disease (MRD)[4–7]. In addition, they have been mechanistically associated with chemoresistance and, in keeping with this, are enriched at relapse[4,8–12]. CREBBP mutations have also been described as an adverse prognostic factor in ALL, acute myeloid leukemia (AML) and follicular lymphoma[13–16]. CREBBP is a large protein with histone acetyltransferase (HAT) enzymatic activity alongside protein scaffolding function mediated through multiple protein-protein interaction domains, including a bromodomain responsible for binding acetylated lysine residues. Alongside its paralogue EP300, CREBBP is considered to primarily function as a transcriptional co-activator, responsible for acetylating histone residues at gene enhancers and promoters. CREBBP LOF mutations can include complete loss of the protein or recurrent point mutations affecting the HAT domain, which appear to exert a stronger phenotype[4]. During B-ALL evolution, CREBBP mutations frequently become bi-allelic and commonly co-associate with activating RAS pathway mutations, suggesting strong oncogenic co-operativity[4,12,17].

Targeting cells harboring LOF mutations in tumor suppressor genes (TSG) principally relies on perturbing synthetic lethal dependencies acquired upon loss of TSG activity, commonly via inhibition of redundant pathways or protein paralogues. In the context of CREBBP, this has been demonstrated in models of B-cell lymphoma through inhibition of residual EP300 function using small molecule HAT or bromodomain inhibitors[18]. Global analyses of genetic co-dependencies have also implicated a dependency of CREBBP-mutated tumors on EXOC5 function[19], whilst a number of mechanistic studies have identified potentially targetable roles for CREBBP in modulating key cellular processes including DNA damage response, signaling, apoptosis and metabolism[3,12,17,20].

In this study we sought to develop treatment options targeting CREBBP-mutated high-risk B-ALL. We generated isogenic human B-ALL cell lines and undertook a synthetic lethal drug screen focusing on clinically-actionable agents targeting pathways mechanistically associated with CREBBP function. CREBBP LOF resulted in cell cycle and metabolic dysregulation, with prominent changes in lipid metabolism and marked sensitivity to ferroptotic cell death upon small molecule inhibition of the anti-apoptotic regulator BCL2. Inhibition of CREBBP function with small molecule inhibitors could phenocopy this synthetic lethal effect, sensitizing diverse subtypes of B-ALL to BCL2 inhibitors in-vitro and producing a significant survival advantage in-vivo, thus providing a potentially efficacious drug combination across a wider number of B-ALL genotypes.

## Results

### CREBBP-mutated B-ALL cell lines show increased sensitivity to Venetoclax

To identify candidate therapeutics that specifically target CREBBP-mutated high-risk B-ALL, we undertook a synthetic-lethal drug screen. The CREBBP wild-type (WT) B-ALL cell line 697 (derived from a patient with high-risk relapsed E2A::PBX1 B-ALL)[21] was genome-engineered by CRISPR-Cas9 homologous recombination to introduce a recurrent hotspot mutation at arginine 1446 (CREBBP^R1446C), which is implicated to exert a dominant-negative effect on CREBBP acetyltransferase activity[4]. Several clones were generated, including a homozygous CREBBP^R1446C knock-in mutant clone (hereafter, 697^KI) and a mutant clone containing two frameshift mutations resulting in a complete knockout of CREBBP protein (hereafter, 697^KO) (Fig. 1a and Supplementary Fig. 1a). For use in validation studies, the ETV6::RUNX1-driven cell line REH (containing three WT copies of CREBBP) was also edited, resulting in two compound-heterozygous mutated clones, each including a single allele of the CREBBP^R1446C HAT mutation, alongside presumed deleterious mutations of the other two alleles (Supplementary Fig. 1a).

We subjected the 697 isogenic cell lines to a targeted drug screen, using a wide range of concentrations, focussed on clinically-actionable

drugs in classes implicated or hypothesized to show differential sensitivity in published models of B-cell lymphoma and other CREBBP-mutated malignancies (Supplementary Data Table 1)[3,4,10–12,18,20,22]. CREBBP-mutated 697 cells were not differentially sensitive to traditional cytotoxic chemotherapy, and paradoxically showed a degree of sensitization to the glucocorticoid Dexamethasone, used in current ALL induction regimens (Fig. 1b and Supplementary Data Table 1 and Fig. 1b)[4,11,12]. As anticipated, and validating our screen design, inhibitors of CREBBP and its paralogue EP300 (the CREBBP/EP300-specific bromodomain inhibitor Inobrodib and the CREBBP/EP300 acetylase inhibitor A485) exhibited synthetic lethality, consistent with previous reports in B-cell lymphoma (Fig. 1b, c and Supplementary Data Table 1 and Fig. 1c)[18].

Unexpectedly, the most potent hit identified from the screen was the clinical-grade BCL2 inhibitor Venetoclax, which showed a 2-$\log_{10}$-fold reduction in $IC_{50}$, in both 697^KI and 697^KO clones (Fig. 1b, d and Supplementary Data Table 1 and Fig. 1d). These findings were validated in isogenic REH lines, with a 1-$\log_{10}$-fold reduction in $IC_{50}$ in both mutant clones (Supplementary Fig. 1e). We confirmed this sensitization to Venetoclax in in-vitro proliferation assays by direct cell counting (Fig. 1e). Upon low-dose Venetoclax exposure, CREBBP-mutated 697 cells showed enhanced evidence of markers of programmed cell death, including mitochondrial depolarization, externalization of Annexin-V and induction of cleaved PARP and caspase-3 (Fig. 1f–h and Supplementary Fig. 1f), consistent with the known mechanism-of-action of Venetoclax in inducing apoptotic programmed cell death.

Overall, this focussed drug screen demonstrates that CREBBP-mutated B-ALL is: (i) not uniformly chemo-resistant, and (ii) identifies a number of clinically-actionable agents for use in CREBBP-mutated high-risk B-ALL, including Dexamethasone, EP300 inhibitors and a potent sensitization to the BCL2 inhibitor Venetoclax.

### Venetoclax exerts its effect on CREBBP-mutated B-ALL cell lines by on-target inhibition of BCL2

We sought to explore the mechanism-of-action of Venetoclax. Venetoclax was developed to induce apoptosis through inhibition of BCL2 binding to the pro-cell death proteins BAK and BAX. However, recently, it has been shown to have alternative mechanisms-of-action, in particular on metabolism and self-renewal, including potential BCL2-independent effects[23–25]. To test whether the mechanism-of-action of Venetoclax in CREBBP-mutated B-ALL was through on-target BCL2 inhibition, we employed a doxycycline-inducible shRNA knockdown system, where shRNA expression was directly linked to a fluorescent reporter (Supplementary Fig. 2a–d)[26].

Co-culture of 697^WT or 697^KI cells expressing one of two unique BCL2-targeting shRNAs (reported by mCherry) competed against cells expressing an shRNA targeting the negative-control renilla gene (reported by green fluorescent protein (GFP)) showed that 697^KI cells exhibited a marked competitive disadvantage upon BCL2-knockdown, when compared to 697^WT (Fig. 2a, b). This was associated with significant externalization of Annexin-V, consistent with the induction of programmed cell death (Fig. 2c, d).

Furthermore, 697^KI cells also showed differential sensitivity to the structurally-unrelated clinical-grade dual BCL2/BCL_XL inhibitor Navitoclax (Fig. 2e and Supplementary Fig. 2e, f). Conversely, specific inhibitors of other antiapoptotic proteins (BCL_XL and MCL1) showed no specificity for CREBBP-mutated cells (Fig. 2f), confirming a BCL2-specific effect. Collectively these studies demonstrate that Venetoclax induces programmed cell death in CREBBP-mutated B-ALL by on-target BCL2 inhibition and that the sensitivity of 697^KI cells is specific to BCL2 and does not occur through other anti-apoptotic proteins.

### CREBBP-mutated B-ALL cell lines show significant cell cycle and metabolic dysregulation

To further explore the mechanism of action of Venetoclax in CREBBP-mutated 697 cells, we undertook bulk RNA sequencing (RNAseq) of

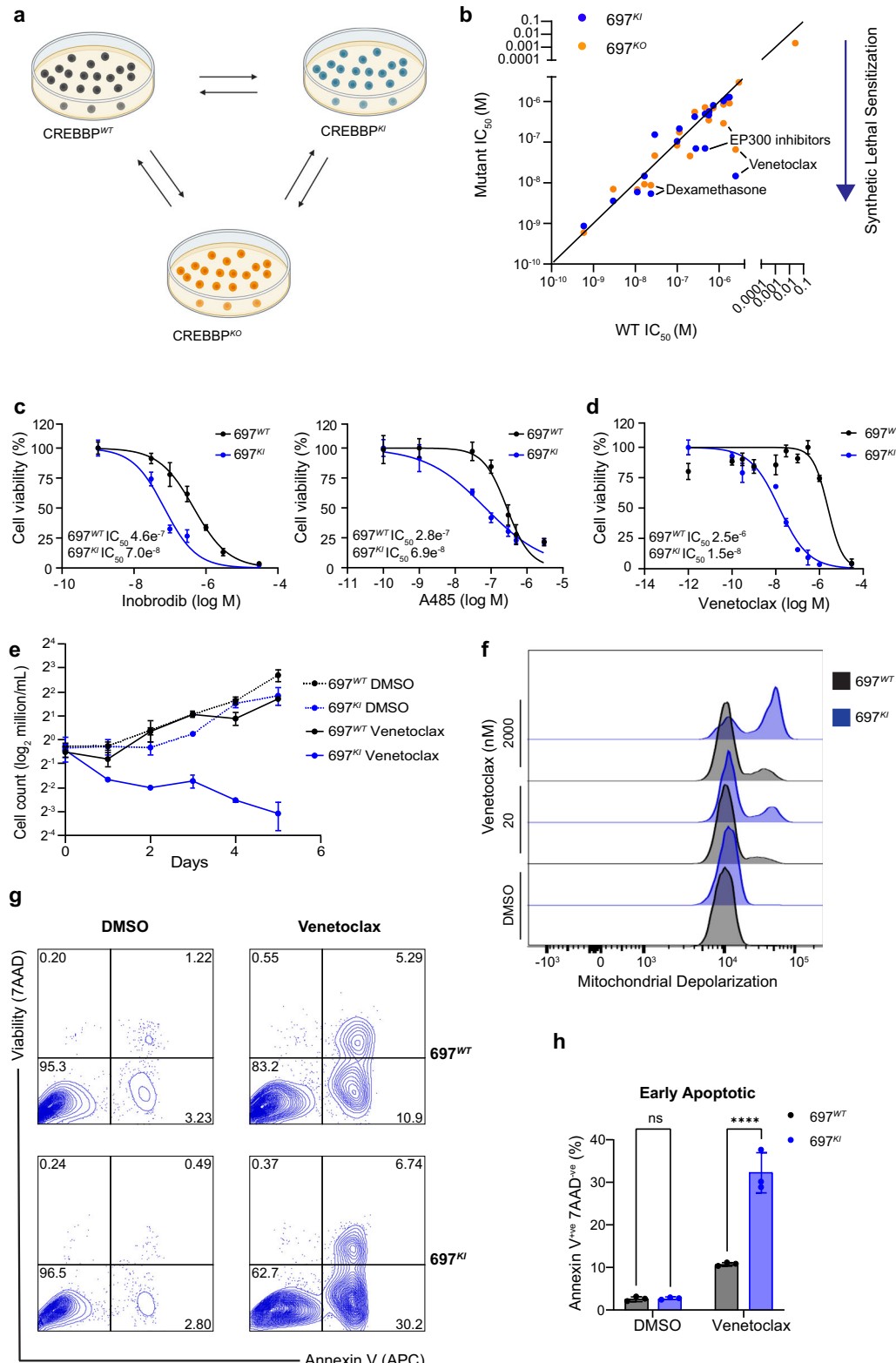

697^{WT} and 697^{KI} cells, after 24 h exposure to either dimethyl sulfoxide (DMSO) vehicle, or low-dose Venetoclax (20 nM—the IC$_{50}$ of 697^{KI} cells). Consistent with the role of *CREBBP* as a transcriptional co-activator, the majority of differentially expressed genes (DEGs) between DMSO-vehicle treated 697^{WT} and 697^{KI} cells were down-regulated (Supplementary Fig. 3a). Gene Set Enrichment Analysis (GSEA) also showed marked down-regulation of published *Crebbp* target genes from a mouse lymphoma model (Supplementary Fig. 3b)[27].

We explored this further using chromatin immunoprecipitation of H3K27ac marks alongside CUT&RUN assessment of CREBBP binding sites. 697^{KI} showed a reduction in the total number of both CREBBP-bound loci and H3K27ac modified peaks (Supplementary Fig. 3c). Furthermore, the normalized read enrichment of CREBBP binding at transcriptional start sites (TSSs) was also consistently reduced in 697^{KI} vs. 697^{WT}, irrespective of differential gene expression, whereas H3K27ac was predominantly lost at TSSs of genes that were

**Fig. 1 | CREBBP-mutated B-ALL cell lines show increased sensitivity to Venetoclax. a** Schematic of genome editing strategy used to engineer $697^{WT}$ (black) into isogenic *CREBBP* $697^{KI}$ (blue) and $697^{KO}$ (yellow) clones. Created in BioRender. Huntly, B. (2025) https://BioRender.com/s71p510. **b** Cell-based drug screening was performed using a panel of 32 small molecule compounds predicted to have differential sensitivity between *CREBBP* WT and mutant lines. 72 h viability was measured by MTS assays using a wide concentration range between 3pM and 30 μM in $n = 3$ technical replicates and repeated using narrower concentration ranges to define $IC_{50}$ where appropriate. Results are presented as an $IC_{50}$ ratio of $697^{KI}$ (blue) and $697^{KO}$ (yellow) clones compared to WT. **c** Dose response curves of two CREBBP/EP300 inhibitors Inobrodib (left) and A485 (right) showing enhanced sensitivity of $697^{KI}$ (blue) compared to $697^{WT}$ (black) in 72 h MTS viability assay. $n = 3$ technical replicates, mean ± SD. **d** Dose response curve of $697^{WT}$ (black) and $697^{KI}$ (blue) lines to Venetoclax in 72 h MTS viability assay. $n = 3$ technical replicates, mean ± SD. **e** Growth curve of $697^{WT}$ (black) and $697^{KI}$ (blue) grown in the presence of either DMSO vehicle (dotted lines) or 20 nM Venetoclax (solid lines). $n = 3$ independent replicates, mean ± SD. **f** Mitochondrial depolarization as assessed by staining for JC1 by flow cytometry (488 nm 530/30) in response to DMSO vehicle, or Venetoclax at 20 or 2000nM in $697^{WT}$ (black) and $697^{KI}$ (blue) cell lines. **g** Representative flow cytometry plots of externalization of Annexin-V (reported by APC) in response to 24 h exposure to DMSO vehicle (left) or Venetoclax 20 nM (right) in $697^{WT}$ (top) and $697^{KI}$ (bottom) cell lines. Viability is assessed by 7AAD exclusion. **h** Proportion of viable 7AAD$^{-ve}$Annexin-V$^{+ve}$ early apoptotic cells. $n = 3$ independent replicates. Mean ± SD, 2-way ANOVA ****, $P = 0.000009$. Source data are provided as a Source Data file.

---

downregulated in $697^{KI}$ (Fig. 3a). Binding and Expression Target Analysis (BETA) linking differential gene expression to CREBBP-bound enhancers confirmed a significant enrichment for gene down-regulation near CREBBP-bound enhancers in $697^{KI}$, with a corresponding up-regulation of DEGs related to CREBBP-bound enhancers in $697^{WT}$ (Fig. 3b)[28]. Collectively, these findings are consistent with the proposed mechanism of the $CREBBP^{R1446C}$ mutation as detrimental to CREBBP co-activator function[4].

KEGG pathway and GSEA analysis of DEGs showed significant down-regulation of signatures associated with apoptosis, in consonance with our functional experiments above (Fig. 1f–h and Fig. 3c, d). However, gene-specific examination of differential expression of apoptotic regulators by KEGG pathway analysis showed a mixed picture, affecting the expression of both pro- and anti-apoptotic genes, including a small but significant up-regulation of *BCL2* itself (Fig. 3e and Supplementary Fig. 3d).

More broadly, KEGG and GSEA pathway analyses showed a differential down-regulation of cell cycle and signaling pathways in $697^{KI}$ (Fig. 3c, d and Supplementary Fig. 3e). We confirmed a relative reduction in proliferative capacity in both $697^{KI}$ and $697^{KO}$ cells compared to $697^{WT}$ by proliferation assays (Fig. 3f). This was associated with a significantly increased proportion of $697^{KI}$ cells in G1 phase alongside reduced Early S/G2-S-M phases (Fig. 3g and Supplementary Fig. 3f) confirming a significant defect in cell cycle progression. Down-regulation of cell cycle-associated transcriptional signatures was associated with a marked up-regulation of the tumor suppressor *CDKN2A* (encoding the negative cell-cycle regulator P16$^{INK4a-ARF}$), which is commonly mutated in B-ALL (Fig. 3h). Consistent with this, we observed limited differences in proliferation in isogenic *CREBBP*-mutated REH cell lines, which carry biallelic loss of *CDKN2A* (Supplementary Fig. 3g).

The majority of transcriptionally up-regulated KEGG and Gene Ontology (GO) pathways were indicative of metabolic dysfunction, and GSEA showed dysregulation of fatty acid metabolism and hypoxic gene signatures (Fig. 3c, d and Supplementary Fig. 3h). Given the close association of cell cycle and metabolism, and the established role of BCL2 and Venetoclax in disturbing mitochondrial respiration[25], we analysed baseline metabolic differences between $697^{WT}$ and $697^{KI}$ cells using in-vitro metabolic flux assays. Unexpectedly, and despite lower cell cycle progression, $697^{KI}$ cells consistently showed increased rates of both glycolysis and oxidative phosphorylation (OxPhos), including higher rates of both basal and maximal respiration, and an increase in spare respiratory capacity (SRC) (Fig. 3i, j; Supplementary Fig. 3i, j).

Collectively our model suggests that loss of CREBBP acetyl-transferase function results in significant transcriptional dysregulation, affecting multiple cellular processes including apoptosis, cell cycle and metabolism.

## Venetoclax induces ferroptotic cell death in *CREBBP*-mutated B-ALL cell lines

To explore the transcriptional impact of low-dose Venetoclax treatment of $697^{KI}$ cells, we employed a four-way interaction model to identify genes specifically dysregulated in $697^{KI}$ cells upon Venetoclax treatment (Supplementary Fig. 4a). KEGG pathway analyses of these genes showed further down-regulation of cell cycle-associated genes and enrichment for metabolic pathways and ferroptosis in Venetoclax-treated $697^{KI}$ cells (Fig. 4a). Furthermore, GSEA showed marked up-regulation of genes associated with multiple metabolic processes, including ROS scavenging, ferroptosis and the unfolded protein response (Fig. 4b).

Ferroptosis is a distinct form of programmed cell death resulting from iron-catalyzed reactive oxygen species (ROS)-mediated damage to unsaturated fatty acids of membrane phospholipids[29]. It commonly co-associates with apoptosis and is associated with expression of cell death markers, including Annexin-V externalization[30]. We demonstrated evidence of ferroptosis upon Venetoclax treatment, specifically occurring in $697^{KI}$ cells using in-vitro assays. Exposure to the cell-permeable pan-caspase inhibitor Z-VAD partially rescued viability to high-dose Venetoclax in both $697^{WT}$ and $697^{KI}$ cells, indicating a role for intrinsic, caspase-mediated apoptosis induced by high-dose Venetoclax in both lines (Fig. 4c). Conversely, exposure to Liproxstatin 1, which specifically inhibits ferroptosis-associated lipid peroxidation, showed significant rescue only in $697^{KI}$ cells, consistent with ferroptosis being the predominant mechanism of cell death mediating sensitization to low dose BCL2 inhibition (Fig. 4d). BCL2 inhibition with either Venetoclax or Navitoclax was also associated with ferroptosis in $697^{KI/KO}$ and REH$^{Mut}$ cells, as shown by elevated BODIPYC$_{11}$ staining, an indicator of lipid peroxidation (Fig. 4e and Supplementary Fig. 4b–d). Metabolically, low dose (20 nM) Venetoclax specifically resulted in a reduction in both basal and maximal OxPhos only in $697^{KI}$ cells, with no effect seen in $697^{WT}$. In contrast, high-dose Venetoclax (2000nM) resulted in a marked reduction in OxPhos in both $697^{KI}$ and $697^{WT}$ cells, consistent with the role of BCL2 in regulating mitochondrial outer membrane permeabilization (Supplementary Fig. 4e).

Overall, these findings demonstrate that the major driver underlying the sensitivity of *CREBBP*-mutated $697^{KI}$ cells to BCL2 inhibition at low doses is ferroptotic programmed cell death, associated with underlying metabolic dysregulation.

## CREBBP-mutation affects the redox balance and lipid content of B-ALL cell lines

To better understand the mechanisms underlying ferroptosis susceptibility, we tested the intrinsic susceptibility of $697^{WT}$ and $697^{KI}$ cells to ferroptosis using direct inducers of Reduction-Oxidation (redox) stress. $697^{KI}$ cells were slightly more susceptible to induction of ferroptosis by Erastin, which induces ferroptosis in RAS-mutated tumors, by both inhibiting system xc$^-$ (a cystine/glutamate antiporter upstream of glutathione production) and the VDAC family of mitochondrial outer membrane anion channels (Fig. 5a)[31,32]. Similarly, inhibition of glutathione phospholipid peroxidase activity using the GPX4 inhibitor, RSL3, induced significantly higher levels of lipid peroxidation in $697^{KI}$ cells, further demonstrating their enhanced sensitivity to redox stress (Fig. 5b).

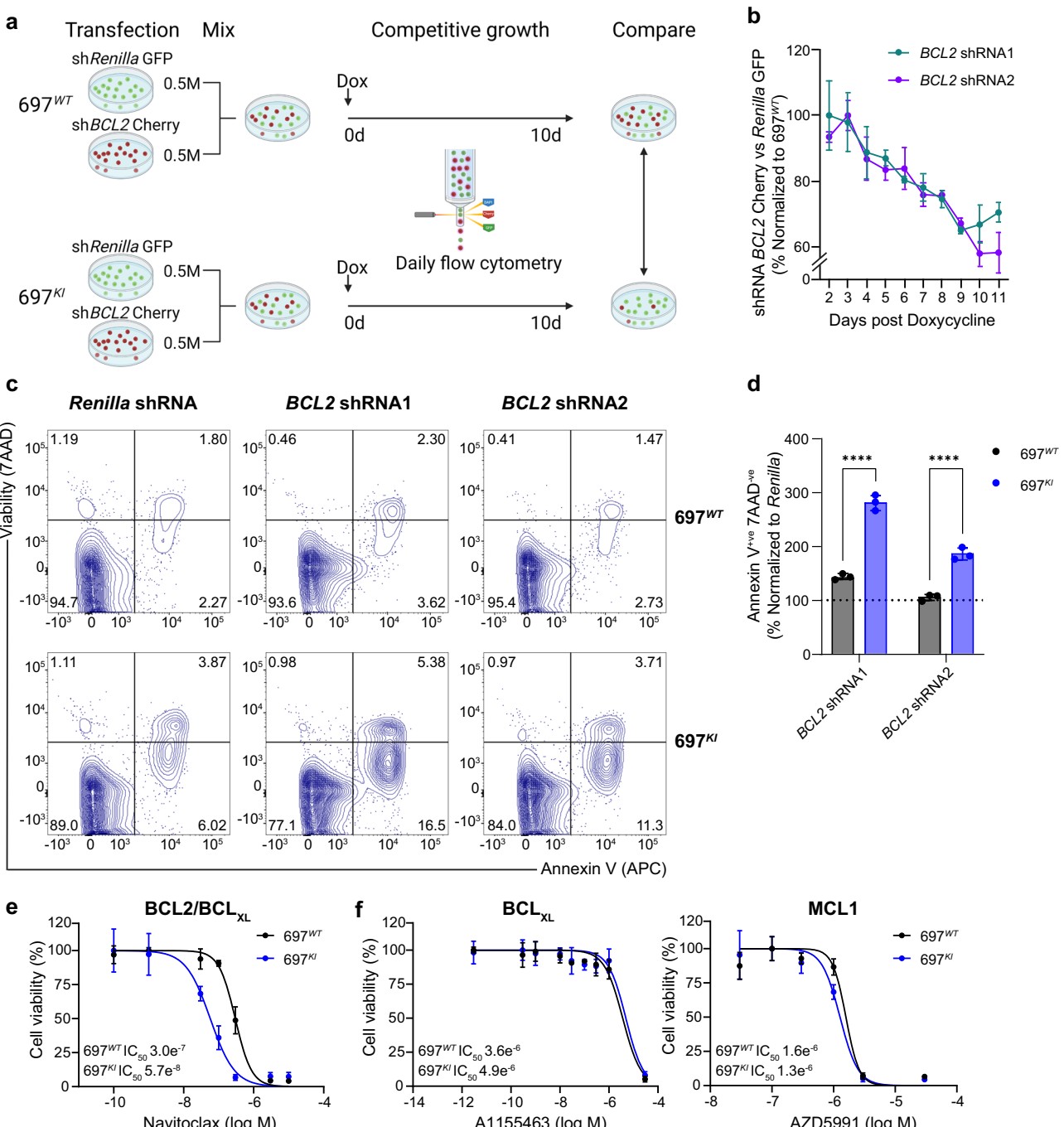

**Fig. 2 | Venetoclax exerts its effect on *CREBBP*-mutated B-ALL cell lines by on-target inhibition of BCL2. a** Schematic of doxycycline-inducible shRNA KD system competitive co-culture assay. $697^{WT}$ and $697^{KI}$ cells were stably transfected with two separate doxycycline-inducible shRNAs targeting *BCL2*, reported by mCherry, or a control shRNA targeting *Renilla*, reported by GFP. *BCL2* and *Renilla* shRNA-expressing cells were mixed in equal numbers, doxycycline added to the media (500 ng/ml) to induce shRNA expression and the proportion of cells surviving *BCL2/Renilla*-KD analysed by daily flow cytometry. Created in BioRender. Huntly, B. (2025) https://BioRender.com/w63h764. **b** *BCL2* shRNA KD competitive proliferation assay for two different *BCL2*-targeting shRNAs are presented, showing the ratio of *BCL2*-targeting shRNA (mCherry) vs. *Renilla* control (GFP), in $697^{KI}$ normalized to $697^{WT}$ as a percentage. $n = 3$ independent replicates, mean ± SD. **c** Representative flow cytometry plots of externalization of Annexin-V (APC) in $697^{WT}$ (top) and $697^{KI}$ (bottom) cell lines in response to doxycycline-induced shRNAs targeting *Renilla* (left), or two different *BCL2*-targeting shRNAs (middle and right). Day 6 post induction. Viability is assessed by 7AAD exclusion. **d** Proportion of viable 7AAD$^{-ve}$Annexin-V$^{+ve}$ early apoptotic cells normalized to doxycycline-induced *Renilla* control. $n = 3$ independent replicates analysed 6 days after induction, mean ± SD, 2-way ANOVA $^{****}$, shRNA1 $P = 0.0000003$ and shRNA2 $P = 0.00002$. **e** Dose response curve of $697^{WT}$ (black) and $697^{KI}$ (blue) to Navitoclax in 72 h MTS viability assays. $n = 3$ technical replicates, mean ± SD. **f** Dose response curve of $697^{WT}$ (black) and $697^{KI}$ (blue) to A1155463 (BCL$_{XL}$ only inhibitor, left) and AZD5991 (MCL1 inhibitor, right) in 72 h MTS viability assays. $n = 3$ technical replicates, mean ± SD. Source data are provided as a Source Data file.

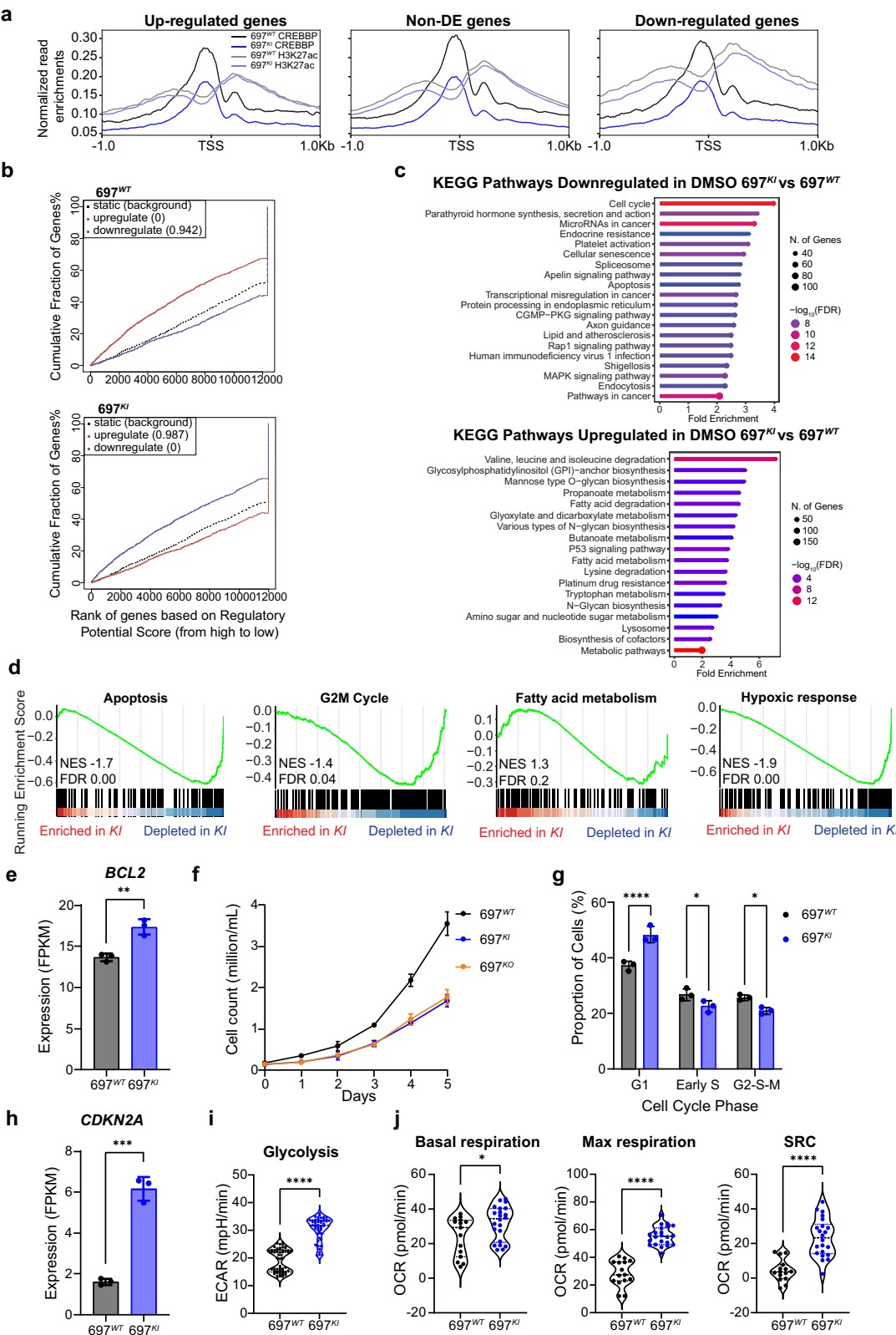

GPX4 is a key inhibitor of lipid peroxidation and has been identified as a recurrent genetic dependency in CRISPR screening of B-ALL cell lines[33]. However, despite the differential sensitivity of $697^{KI}$ cells to the GPX4 inhibitor RSL3, no differences in GPX4 protein abundance were detectable by western blot analysis (Supplementary Fig. 5a). To investigate the abundance of ROS scavenging proteins more broadly, we performed total proteomics on $697^{WT}$ and $697^{KI}$ cells. To our surprise, $697^{KI}$ cells were characterized by significant up-regulation of proteins in multiple antioxidant pathways, including GSS, GSTM2, SPR, SOD1, CAT and TXNRD2 (Fig. 5c; Supplementary Data Table 2), consistent with a compensatory adaptation to ROS-mediated stress. Concordantly, western blot analysis of 4-hydroxynonenal (4HNE) protein adducts, formed in response to lipid peroxidation, indicated lower baseline signal in $697^{KI}$ cells compared to $697^{WT}$. However,

**Fig. 3 | *CREBBP*-mutated B-ALL cell lines show significant cell cycle and metabolic dysregulation. a** Normalized read enrichments of CREBBP binding (dark) and H3K27ac marks (light) in 697$^{WT}$ (black) and 697$^{KI}$ (blue) centered on transcriptional start sites (TSSs), separated on up-regulated (left), non-differentially expressed (center) and down-regulated genes (right). **b** BETA analysis of association of CREBBP-bound H3K27-acetylated enhancers with gene targets, showing association for active differential gene expression in 697$^{WT}$ (top) and repressive differential gene expression in 697$^{KI}$ (bottom). Reported *P* values shown in legend. *P* values comparing the UP/DOWN and non-differentially expressed gene sets calculated using the Kolmogorov-Smirnov test. **c** Summary of most significantly down-regulated (left) and up-regulated (right) KEGG pathways from DEGs of RNAseq analysis comparing DMSO vehicle-treated 697$^{KI}$ with 697$^{WT}$. **d** GSEA analysis of ranked genes of RNAseq analysis comparing DMSO vehicle-treated 697$^{KI}$ with 697$^{WT}$. NES normalized enrichment score, FDR false discovery rate. **e** Comparison of *BCL2* expression in 697$^{WT}$ (black) and 697$^{KI}$ (blue) cells by RNAseq. Each dot represents one sample. Bars show mean fragments per kilobase of transcript per million fragments mapped (FPKM) value ± SD, *n* = 3 independent replicates, significance calculated by two-tailed unpaired *t* test, **P = 0.0038. **f** Proliferation of untreated 697$^{WT}$ (black), 697$^{KI}$ (blue) and 697$^{KO}$ (yellow) cells measured by direct counting. *n* = 3 independent replicates, mean ± SD. **g** Analysis of cell cycle stage by FUCCI reporter system. Percentage cells in G1, Early S and G2-S-M phases in 697$^{WT}$ (black) and 697$^{KI}$ (blue) cells. *n* = 3 independent replicates, bar shows mean average ± SD; significance calculated by 2-way ANOVA, ****P = 0.00004, Early S P = 0.0498; G2-S-M P = 0.0342. **h** Comparison of *CDKN2A* expression in 697$^{WT}$ (black) and 697$^{KI}$ (blue) cells by RNAseq. Each dot represents one sample. Bars show mean FPKM value ± SD, *n* = 3 independent replicates, significance calculated by two-tailed unpaired *t* test, ***P = 0.0002. **i** Glycolytic rate measured by extracellular acidification rate (ECAR) in 697$^{WT}$ (black) and 697$^{KI}$ (blue) cells. Summary of maximal ECAR. Each dot represents a single replicate acquired from two separate experiments. Significance calculated by two-tailed unpaired *t* test, ****P < 1 × 10$^{-15}$. **j** Mitochondrial oxygen consumption rate (OCR). Summary of basal OCR (left), maximal OCR (middle) and spare respiratory capacity (SRC) (right). Each dot represents a single replicate acquired from two separate experiments. Significance calculated by two-tailed unpaired *t* test, *, P = 0.0315; ****, Maximal respiration P = 6.2 × 10$^{-12}$, SRC P = 7.3 × 10$^{-7}$. Source data are provided as a Source Data file.

whereas these adducts reduced in response to low dose Venetoclax in 697$^{WT}$, they actually increased in 697$^{KI}$, consistent with a differential ability between the isogenic cell lines to tolerate ferroptotic stress (Supplementary Fig. 5b).

Total proteomics also showed significant dysregulation of genes implicated in ferroptosis sensitivity, including down-regulation of VDAC1 and multiple enzymes involved in the synthesis of ether-linked lipids (AGPS, GNPAT, ACSL4) in the 697$^{KI}$ cell line (Fig. 5c; Supplementary Data Table 2). To assess potential alterations in lipid composition, we therefore performed comprehensive quantitative lipidomic analysis, using high-resolution mass spectrometry of 697$^{WT}$ and 697$^{KI}$ cell lines. Whilst the total levels of major structural membrane lipids were similar between the cell lines (Supplementary Data Table 3), 697$^{KI}$ cells had markedly higher levels of polyunsaturated fatty acids (PUFAs) in their structural lipid species, which are more vulnerable to ROS damage and lipid peroxidation (Fig. 5d; Supplementary Fig. 5c). Furthermore, consistent with the transcriptional down-regulation of *AGPS*, and low protein levels of GNPAT and ACSL4, 697$^{KI}$ cells had very low levels of ether-linked lipids: alkyl-acylphospholipids and alkenyl-acylphospholipids (plasmalogens) (Fig. 5e). Ether-linked lipids are synthesized in peroxisomes and the endoplasmic reticulum, and play key signaling roles in these compartments, as well as in the regulation of mitochondrial redox control, including the regulation of ROS balance and PUFA storage[34,35], and their concentration is inversely correlated to ferroptosis sensitivity in different models[35–37].

Lastly, we sought to cross-correlate the findings from our cell line models with transcriptional analysis from the large Phase 2 TARGET patient cohort[38]. Only a small number of patients were identified to carry *CREBBP* mutations, therefore to more holistically link CREBBP-activity to ferroptotic susceptibility, we correlated gene sets involved in ferroptosis with *CREBBP* expression, demonstrating a highly significant negative correlation ($R = -0.42$, p = 6.6e$^{-10}$) (Supplementary Data Table 4, Fig. 5f). Consistent with the total proteomic analysis in Fig. 5c, low-*CREBBP*-expressing patients demonstrated upregulation of ROS scavenger genes, including *GPX1, TXN2* and *SOD1*, peroxisomal/ether lipid regulators, including *PEX7* and *FAR2*, and relative downregulation of anti-ferroptotic regulators such as *ASCL3* (Fig. 5g).

Overall, these findings demonstrate that *CREBBP*-mutated 697$^{KI}$ cells exhibit up-regulation of ROS scavenging pathways, increased polyunsaturation of structural lipids and depletion of ether-linked signaling lipids. The analysis of patient transcriptional data corroborates that low *CREBBP*-expression correlates with these signatures, suggesting this may be conserved in multiple subtypes of B-ALL.

## Acquisition of Venetoclax resistance results in transcriptional convergence

Treating cancer with single agents almost inevitably results in acquired treatment resistance, which can often be mitigated by combination therapies. Understanding resistance pathways can: (i) provide insight into pharmacological mechanism-of-action; (ii) identify agents effective in relapse and; (iii) identify novel sensitivities to prevent resistance and improve primary treatment efficacy.

Venetoclax-resistant 697$^{WT\text{-Res}}$ and 697$^{KI\text{-Res}}$ cells were generated in parallel, by escalated exposure to Venetoclax, to generate lines capable of proliferating in 2000nM Venetoclax (the IC$_{50}$ dose of 697$^{WT}$) (Supplementary Fig. 6a). Resistant cells were transcriptionally compared to sensitive parental lines by bulk RNAseq. Global profiling by principal component analysis showed the major transcriptional variance separating resistant from sensitive lines (PC1; 77% variance) (Fig. 6a). Venetoclax resistance resulted in a high degree of transcriptional concordance between resistant cell lines, with 697$^{WT\text{-Res}}$/697$^{KI\text{-Res}}$ cells differing by only 31 (LFC > 1) (Supplementary Fig. 6b) compared to 559 DEGs between 697$^{WT}$/697$^{KI}$. Notably, the most differentially down-regulated genes in 697$^{KI\text{-Res}}$/697$^{WT\text{-Res}}$ were the redox-sensing selenoprotein *SELENON* and the ether-linked lipid biosynthetic enzyme and ferroptosis regulator *AGPS*[39], one of the most transcriptionally down-regulated genes in 697$^{KI}$ vs. 697$^{WT}$, and which is derepressed in 697$^{KI\text{-Res}}$ (Supplementary Fig. 6c). Functionally, 697$^{KI\text{-Res}}$ cells also showed complete reversal of their sensitization to lipid peroxidation upon GPX4 inhibition with RSL3 (Fig. 6b).

Both 697$^{WT\text{-Res}}$ and 697$^{KI\text{-Res}}$ cells significantly up-regulated *BCL2* (Fig. 6c), but showed no consistent changes in *MCL1* or *BCL$_{XL}$* (Supplementary Fig. 6d). GSEA showed prominent up-regulation of cell cycle signatures in both resistant lines, which was more marked in 697$^{KI\text{-Res}}$ cells compared to 697$^{WT\text{-Res}}$ (Fig. 6d). This was associated with down-regulation of *CDKN1A* (encoding the cellular senescence regulator P21) in both lines and a specific down-regulation of *CDKN2A* in the 697$^{KI\text{-Res}}$ cells[40] (Fig. 6e).

Metabolic signatures were prominently up-regulated in both resistant cells, although distinct changes were seen for OxPhos, hypoxia and RAS pathway signatures (Supplementary Fig. 6e). Functional metabolic profiling demonstrated reduced basal and maximal respiration upon resistance in both lines (Fig. 6f and Supplementary Fig. 6f). Conversely, glycolytic rate increased upon resistance in 697$^{WT}$ cells, whereas it was markedly reduced in 697$^{KI}$ cells, indicating differential metabolic reprogramming upon Venetoclax resistance (Fig. 6g and Supplementary Fig. 6g).

Collectively, these findings identify multiple adaptations upon Venetoclax resistance that were shared by 697$^{WT\text{-Res}}$ and 697$^{KI\text{-Res}}$ cells, including transcriptional up-regulation of *BCL2*, a reduction in the rate

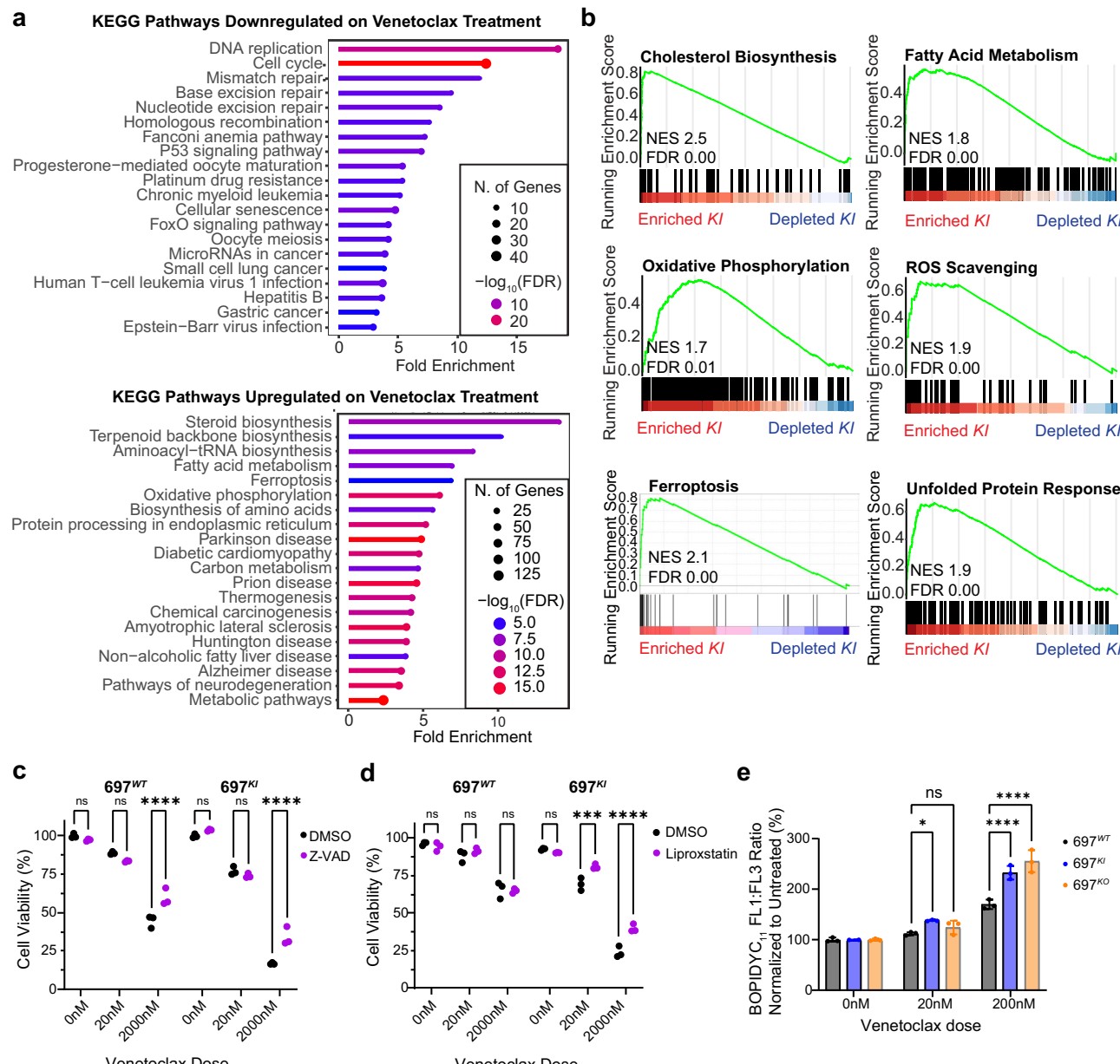

**Fig. 4 | Venetoclax induces ferroptotic cell death in *CREBBP*-mutated B-ALL cell lines. a** Summary of most significantly down-regulated (top) and up-regulated (bottom) KEGG pathways from DEGs of RNAseq analysis comparing Venetoclax-treated 697$^{KI}$ with 697$^{WT}$. **b** GSEA analysis of ranked genes of RNAseq analysis comparing Venetoclax-treated 697$^{KI}$ with 697$^{WT}$. **c** Cellular viability assessed by flow cytometry of 697$^{WT}$ (left) and 697$^{KI}$ (right) cells treated with increasing doses of Venetoclax at 24 h either with exposure to the cell-permeant pan-caspase inhibitor Z-VAD (purple) or DMSO control (black). $n = 3$ independent replicates, analysed by 2-way ANOVA, ****, 697$^{WT}$ $P = 0.000006$; 697$^{KI}$ $P = 0.0000008$. **d** Cellular viability assessed by flow cytometry of 697$^{WT}$ (left) and 697$^{KI}$ (right) cells treated with

increasing doses of Venetoclax at 24 h, either with exposure to the ferroptotic inhibitor Liproxstatin 1 (purple) or DMSO control (black). $n = 3$ independent replicates, analysed by 2-way ANOVA; ***$P = 0.0004$; ****$P = 0.000004$. **e** Lipid peroxidation in Venetoclax-treated 697$^{WT}$ (black), 697$^{KI}$ (blue) and 697$^{KO}$ (yellow) cells assessed by BODIPYC$_{11}$ expressed as a ratio of FL1:FL3 (488 nm 530/30:610/20) mean fluorescence intensity (MFI) normalized to untreated cells. n = 3 independent replicates, each dot represents a single sample, mean ± SD, two-way ANOVA, ****, 697$^{WT}$ vs. 697$^{KI}$,$P = 0.000003$, 697$^{WT}$ vs. 697$^{KO}$, $P = 4 \times 10^{-4}$; *$P = 0.0179$. Source data are provided as a Source Data file.

of oxidative phosphorylation and transcriptional upregulation of *VDAC1*, alongside the reversal of specific phenotypes acquired upon *CREBBP* loss in 697$^{KI}$ cells, including normalization of *CDKN2A* expression[40] and glycolytic rate, and re-expression of the plasmalogen lipid regulator *AGPS*.

**Acquisition of Venetoclax resistance sensitizes cells to Erastin**

KEGG pathway analysis showed significant transcriptional changes in ferroptotic and cellular senescence pathways specifically in 697$^{KI-Res}$

cells, driven by down-regulation of *P16$^{INK4A-ARF}$* and up-regulation of multiple *VDAC* genes, notably *VDAC1* (Fig. 7a and Supplementary Fig. 7a, b). VDAC proteins are macro-molecular anion channels located in the outer mitochondrial membrane that co-ordinate apoptosis, ferroptosis, mitochondrial metabolism and directly interact with BCL2[41]. The ferroptosis inducer Erastin works in part by inhibition of VDAC members, including VDAC1[31,42]. In contrast to RSL3 (Fig. 6b), we found a marked induction of Erastin sensitivity upon acquisition of Venetoclax resistance in both resistant cell lines (Fig. 7b). Moreover,

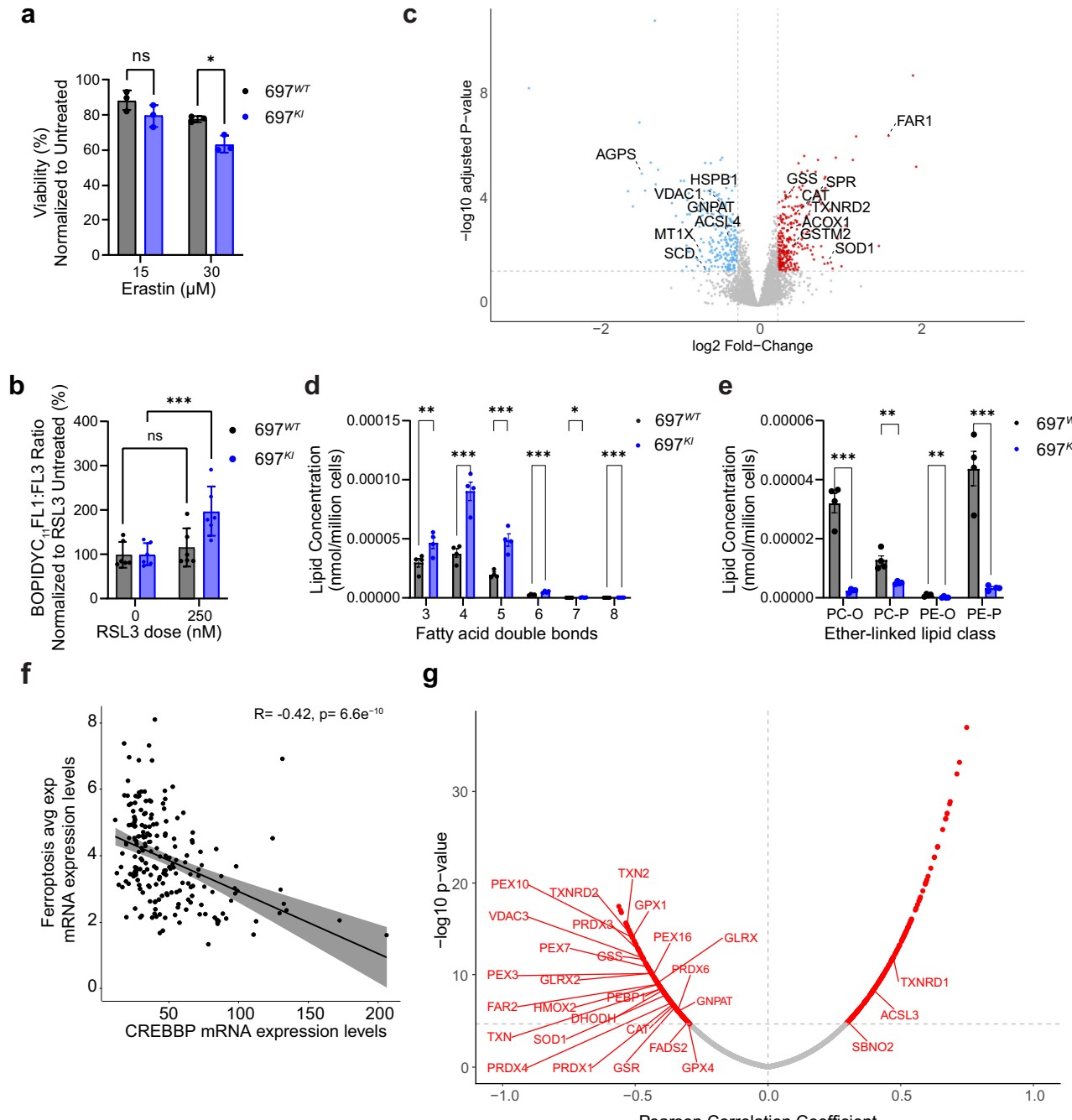

**Fig. 5 | CREBBP-mutation affects the redox balance and lipid content of B-ALL cell lines. a** Viability of $697^{WT}$ (black) and $697^{KI}$ (blue) cells following 48 h exposure to Erastin at 15 or 30 μM. Viability assessed by 7AAD exclusion using flow cytometry. $n = 3$ independent replicates, mean ± SD. Two-way ANOVA; *, $P = 0.014$. **b** Lipid peroxidation in RSL3-treated $697^{WT}$ (black) and $697^{KI}$ (blue) cells assessed by BODIPYC$_{11}$ expressed as a ratio of FL1:FL3 MFI normalized to untreated cells. Mean ± SD $n = 6$ each dot represents an individual passage from two independent experiments performed on separate days. Two-way ANOVA, ****$P = 0.0005$. **c** Volcano plot of total proteomic analysis showing differential protein abundance in $697^{KI}$ vs. $697^{WT}$ (red=increased expression in $697^{KI}$). Two-sided limma statistical test. To control for the false discovery rate (FDR), $p$ values were adjusted using the Benjamini-Hochberg method for multiple testing correction. $n = 4$ independent replicates. **d** Total lipidomic analysis showing the concentration of polyunsaturated structural lipids in $697^{WT}$ (black) vs. $697^{KI}$ (blue) (nmol/million cells). Degree of unsaturation is categorized on the x axis. $n = 4$ independent replicates, mean ± SEM.

* $p$ value < 0.05, **$p$ value < 0.01, ***$p$ value < 0.001. **e** Total lipidomic analysis showing the concentration of ether-linked lipids in $697^{WT}$ (black) vs. $697^{KI}$ (blue) (nmol/million cells). PE-O Alkyl Ether-Linked Phosphatidyl ethanolamine, PE-P Alkenyl Ether-Linked Phosphatidyl ethanolamine (Plasmalogen), PC-O Alkyl Ether-Linked Phosphatidylcholine, PC-P Alkenyl Ether-Linked Phosphatidylcholine (Plasmalogen). $n = 4$ independent replicates, mean ± SEM. * $p$ value < 0.05, **$p$ value < 0.01, ***$p$ value < 0.001. **f** Two-sided Pearson correlation with $p$ value adjusted for multiple testing using Bonferroni correction of gene sets for ferroptosis compared to *CREBBP* expression in patients with RNAseq data in the TARGET Phase 2 cohort. Each dot represents an individual patient ($n = 203$). **g** Two-sided Pearson correlation with $p$ value adjusted for multiple testing using Bonferroni correction of RPKM-normalized gene expression compared to *CREBBP* expression from patients with RNA-seq data in the TARGET Phase 2 cohort ($n = 203$). Significantly correlated genes are highlighted in red ($P_{adj} < 0.05$) with significantly correlated genes related to ferroptosis annotated. Source data are provided as a Source Data file.

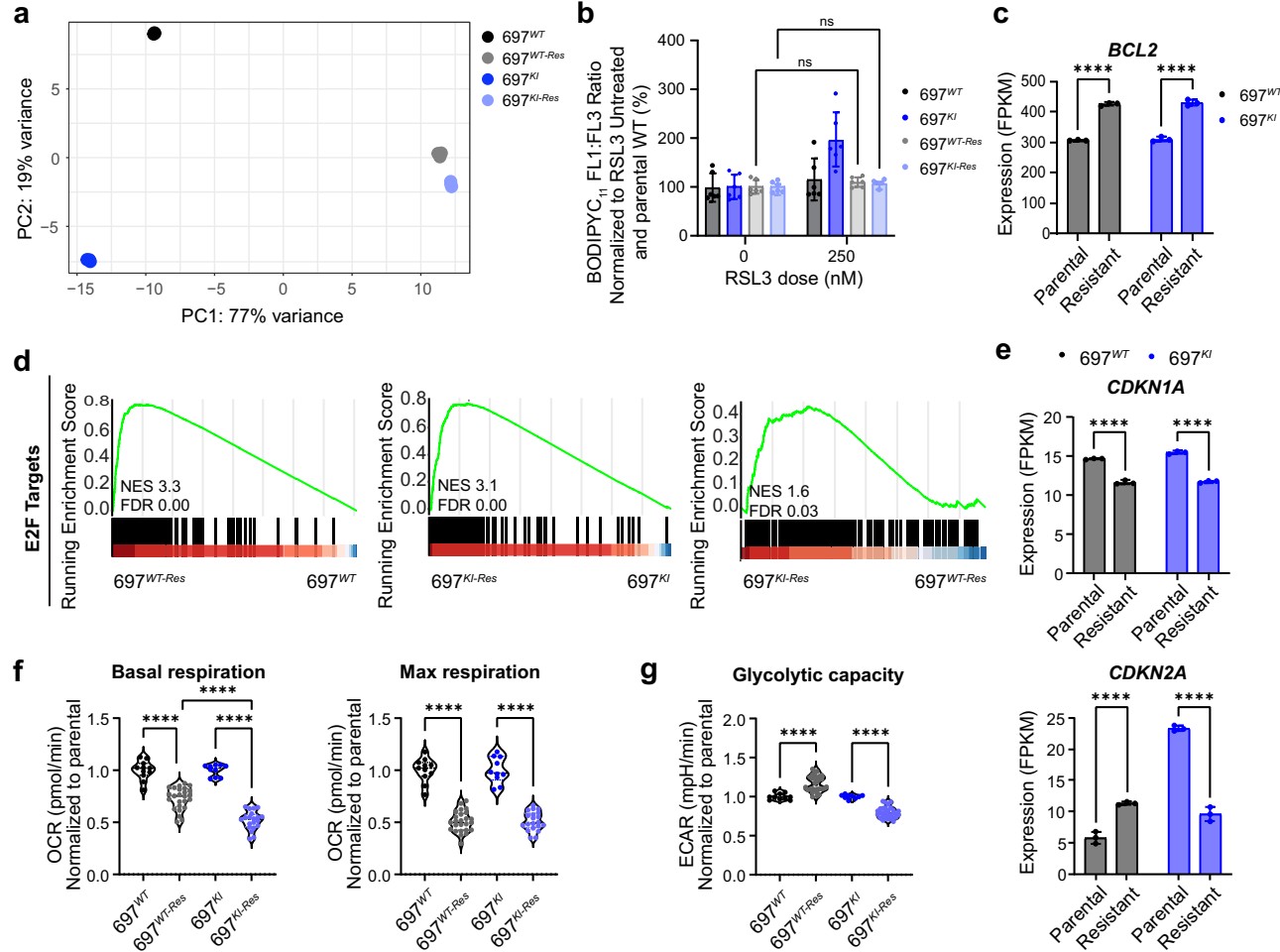

**Fig. 6 | Acquisition of Venetoclax resistance results in transcriptional convergence. a** Principal component analysis of global transcriptional profile by RNAseq of 697^WT (black), 697^KI (blue), 697^WT-Res (gray) and 697^KI-Res (pale blue) cells by RNAseq. $n = 3$ independent replicates, each dot represents one sample. **b** Lipid peroxidation in RSL3-treated 697^WT (black plain), 697^KI (blue plain), 697^WT-Res (black hashed) and 697^KI-Res (blue hashed) cells assessed by BODIPYC_{11} expressed as a ratio of FL1:FL3 MFI normalized to untreated cells and expressed as a percentage of parental 697^WT. Mean ± SD $n = 6$ each dot represents an individual passage from two independent experiments performed on separate days. Two-way ANOVA. **c** Comparison of *BCL2* expression levels in 697^WT (black) and 697^KI (blue) cells comparing parental and Venetoclax-resistant lines. Each dot represents one sample. Bars show mean FPKM value ± SD, $n = 3$ independent replicates. Two-way ANOVA, ****, 697^WT $P = 4.8 \times 10^{-8}$ and 697^KI $P = 4 \times 10^{-8}$. **d** GSEA of ranked RNAseq expression for E2F cell cycle signatures in 697^WT-Res vs. 697^WT (left), 697^KI-Res vs. 697^KI (middle) and 697^KI-Res vs. 697^WT-Res (right). **e** Comparison of *CDKN1A* (left) and

*CDKN2A* (right) expression levels in 697^WT (black) and 697^KI (blue) cells comparing parental and Venetoclax-resistant lines. Each dot represents one sample. Bars show mean FPKM value ± SD, $n = 3$ independent replicates. Two-way ANOVA, ****, left: 697^WT $P = 6 \times 10^{-8}$ and 697^KI $P = 1.1 \times 10^{-8}$; right: 697^WT $P = 0.00006$ and 697^KI $P = 4.9 \times 10^{-8}$. **f** Summary of basal (left) and maximal (right) mitochondrial OCR in 697^WT (black) and 697^KI (blue) cells comparing parental and resistant lines. Each dot represents a single replicate acquired from two separate experiments. One way ANOVA, ****, left: 697^WT vs. 697^WT-Res $P = 2.1 \times 10^{-10}$, 697^WT-Res vs. 697^KI-Res $P = 1.2 \times 10^{-9}$, 697^KI vs. 697^KI-Res $P = 1.9 \times 10^{-11}$; right: 697^WT vs. 697^WT-Res $P = 2 \times 10^{-11}$, 697^KI vs. 697^KI-Res $P = 2 \times 10^{-11}$. **g** Summary of glycolytic rate (ECAR) in 697^WT (black) and 697^KI (blue) cells comparing parental and resistant lines. Each dot represents a single replicate acquired from two separate experiments. One way ANOVA, ****, 697^WT vs. 697^WT-Res $P = 8.7 \times 10^{-7}$, 697^KI vs. 697^KI-Res $P < 1 \times 10^{-15}$. Source data are provided as a Source Data file.

co-exposure of 697^WT, 697^WT-Res and 697^KI-Res cells to increasing doses of Venetoclax and/or Erastin demonstrated pharmacological synergy, indicating a potential route to both overcome and prevent Venetoclax resistance (Fig. 7c).

### Pharmacological inhibition of CREBBP function can sensitize B-ALL cell lines to Venetoclax in-vitro

The majority of B-ALL patients do not have *CREBBP*-mutations. Given the strength of the association with Venetoclax sensitization we hypothesized that pharmacological inhibition of *CREBBP*^WT function could sensitize *CREBBP*^WT B-ALL to BCL2 inhibitors.

We first pre-treated 697^WT or REH^WT cells with the preclinical CREBBP/EP300 HAT inhibitor A485 for 3 days, aiming to generate a stable reduction of CREBBP-mediated acetylation. A485 exposure

almost perfectly phenocopied the Venetoclax sensitization seen in our isogenic lines (Fig. 8a and Supplementary Fig. 8a). Furthermore, co-treatment with A485 and Venetoclax showed strong pharmacological synergy (Fig. 8b). Further corroborating our isogenic findings, A485-sensitized Venetoclax cytotoxicity was associated with enhanced lipid peroxidation, consistent with ferroptosis (Fig. 8c and Supplementary Fig. 8b). Similar results were also seen with the orally-available CREBBP/EP300-specific bromodomain inhibitor Inobrodib, which is currently in clinical trials (Supplementary Fig. 8c,d). In both 697^WT and REH^WT cells, A485-induced HAT inhibition was associated with increased basal and maximal OxPhos with an increase in SRC, similar to that seen in 697^KI cells (Fig. 8d and Supplementary Fig. 8e–g). Single-agent A485-treated 697^WT cells showed no loss of proliferation or evidence of programmed cell death by annexin-V externalization,

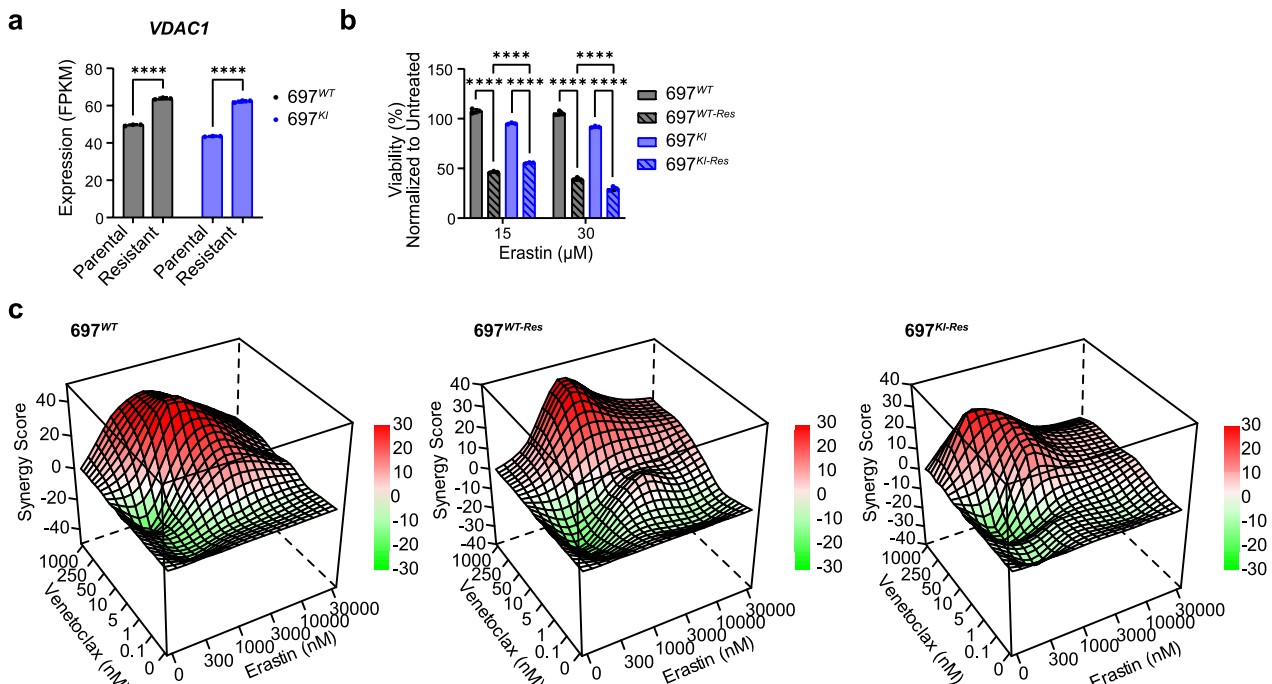

**Fig. 7 | Acquisition of Venetoclax resistance sensitizes cells to Erastin.**
**a** Comparison of *VDAC1* expression levels by RNAseq in 697$^{WT}$ (black) and 697$^{KI}$ (blue) cells comparing parental and Venetoclax-resistant lines. Each dot represents one sample. Bars show mean FPKM value ± SD, *n* = 3 independent replicates. Two-way ANOVA, ****, 697$^{WT}$ *P* = 4.5 × 10$^{-10}$ and 697$^{KI}$ *P* = 5 × 10$^{-11}$. **b** Viability of 697$^{WT}$ (black), 697$^{KI}$ (blue) cells comparing parental (plain) and resistant (hashed) lines following 48 h exposure to Erastin at 15 or 30 μM. Viability assessed by 7AAD exclusion using flow cytometry. *n* = 3 independent replicates, mean ± SD. Two-way

ANOVA, ****, 15 μM: 697$^{WT}$ vs. 697$^{WT-Res}$ *P* = 2.3 × 10$^{-14}$, 697$^{WT-Res}$ vs. 697$^{KI-Res}$ *P* = 7.5 × 10$^{-6}$, 697$^{KI}$ vs. 697$^{KI-Res}$ *P* = 2.4 × 10$^{-14}$; 30 μM: 697$^{WT}$ vs. 697$^{WT-Res}$ *P* = 2.3 × 10$^{-14}$, 697$^{WT-Res}$ vs. 697$^{KI-Res}$ *P* = 3.4 × 10$^{-6}$, 697$^{KI}$ vs. 697$^{KI-Res}$ *P* = 2.3 × 10$^{-14}$. **c** Three-dimensional diffusion plot of ZIP synergy score to combined doses of synchronous Erastin and Venetoclax in 697$^{WT}$ (left) (peak ZIP 41.37), 697$^{WT-Res}$ (middle) (peak ZIP 33.34)and 697$^{KI-Res}$ (right) (peak ZIP 22.69). Viability measured by 72 h MTS assay. Source data are provided as a Source Data file.

strongly suggesting that, at these doses, A485 works by sensitizing cells to Venetoclax-induced cytotoxicity (Fig. 8e, f).

We extended these findings to other human B-ALL cell lines, across a range of genotypes. A485 sensitized the highly Venetoclax-resistant, *BAX*-mutated cell line NALM6 to Venetoclax (Fig. 8g). This was associated with increased lipid peroxidation, indicative of ferroptotic death (Fig. 8h and Supplementary Fig. 8h). Even in Venetoclax-sensitive B-ALL lines driven by high-risk genetic drivers, Venetoclax and A485 showed evidence of further pharmacological synergy (defined as a peak ZIP score ≥10), indicating that this interaction is conserved across diverse genetic subtypes of B-ALL (Fig. 8i).

Collectively, these data pharmacologically validate the findings of BCL2 sensitization identified in our *CREBBP*-mutated cell lines and support a possible drug combination for clinical translation in B-ALL more broadly.

### Genetic or pharmacological inhibition of CREBBP sensitizes B-ALL to Venetoclax in-vivo

Finally, we sought to test whether Venetoclax could target *CREBBP*-mutated B-ALL in-vivo, where cell extrinsic factors and pharmacodynamic effects can result in reduced efficacy or highlight dose-limiting toxicities.

Luciferase-expressing 697$^{WT}$ and 697$^{KI}$ cell lines were engrafted into NOD-SCID-Gamma (NSG) mice (Supplementary Fig. 9a,b). Upon confirmation of engraftment by bioluminescent imaging (BLI), mice were treated with daily oral Venetoclax for up to 30 d (Fig. 9a). As anticipated, Venetoclax exposure was associated with limited disease control in 697$^{WT}$ cells (Fig. 9b, c) resulting in a small, but significant, prolongation of post-transplant overall survival (OS), with all Venetoclax-treated 697$^{WT}$ recipient animals succumbing within 15 d of treatment (median OS 19 vs. 22 d, *p* = 0.0011) (Fig. 9f). 697$^{KI}$ recipient

mice treated with vehicle control succumbed to disease slightly later than 697$^{WT}$, potentially reflecting the proliferation defect characterized in-vitro (median OS 25 vs. 19 d, *p* = 0.0004) (Fig. 9f). Nevertheless, all vehicle-treated 697$^{KI}$ mice succumbed by 19 d of treatment. Venetoclax-treated mice engrafted with 697$^{KI}$ cells demonstrated markedly improved disease control, as assessed by BLI (Fig. 9d, e and Supplementary Fig. 9c, d). Only one Venetoclax-treated 697$^{KI}$ recipient succumbed to disease within the treatment window, with the remaining 5 recipients gaining up to 2 weeks of survival after cessation of Venetoclax exposure (median OS 47 vs. 25 d, *p* = 0.0006) (Fig. 9f).

We next tested whether co-administration of the orally-available, CREBBP bromodomain inhibitor Inobrodib, which is in early-phase trials in hematological malignancies, could sensitize 697$^{WT}$ cells to Venetoclax in-vivo (Supplementary Fig. 9e). Once-daily oral dosing of single-agent Inobrodib did not confer a survival benefit compared to recipients treated with either vehicle or single-agent Venetoclax. In contrast, recipient mice treated with both agents demonstrated a significant survival advantage (median OS combination 34.5 vs. Inobrodib only 24.5 d, *p* = 0.0084) (Fig. 9g).

Finally, we established a PDX from a diagnostic sample obtained from a child with High Hyperdiploid B-ALL, harboring a heterozygous deleterious insertion mutation in *CREBBP*, alongside an *NRAS$^{G12C}$* activating mutation, and who responded poorly to treatment, as evidenced by the presence of high-risk measurable residual disease (2.3%) at the end of induction. Engrafted NSG mice were treated with oral Vehicle control, Inobrodib, Venetoclax, or Combination treatment from 77 days after transplant for three consecutive days per week, up to a maximum of 10 weeks. Despite the deleterious *CREBBP* mutation, Inobrodib monotherapy failed to show a significant OS advantage over Vehicle-treated mice (median OS 120 vs. 114 d, *p* = 0.2) (Fig. 9h). Conversely, Venetoclax showed a highly significant OS advantage (133 vs.

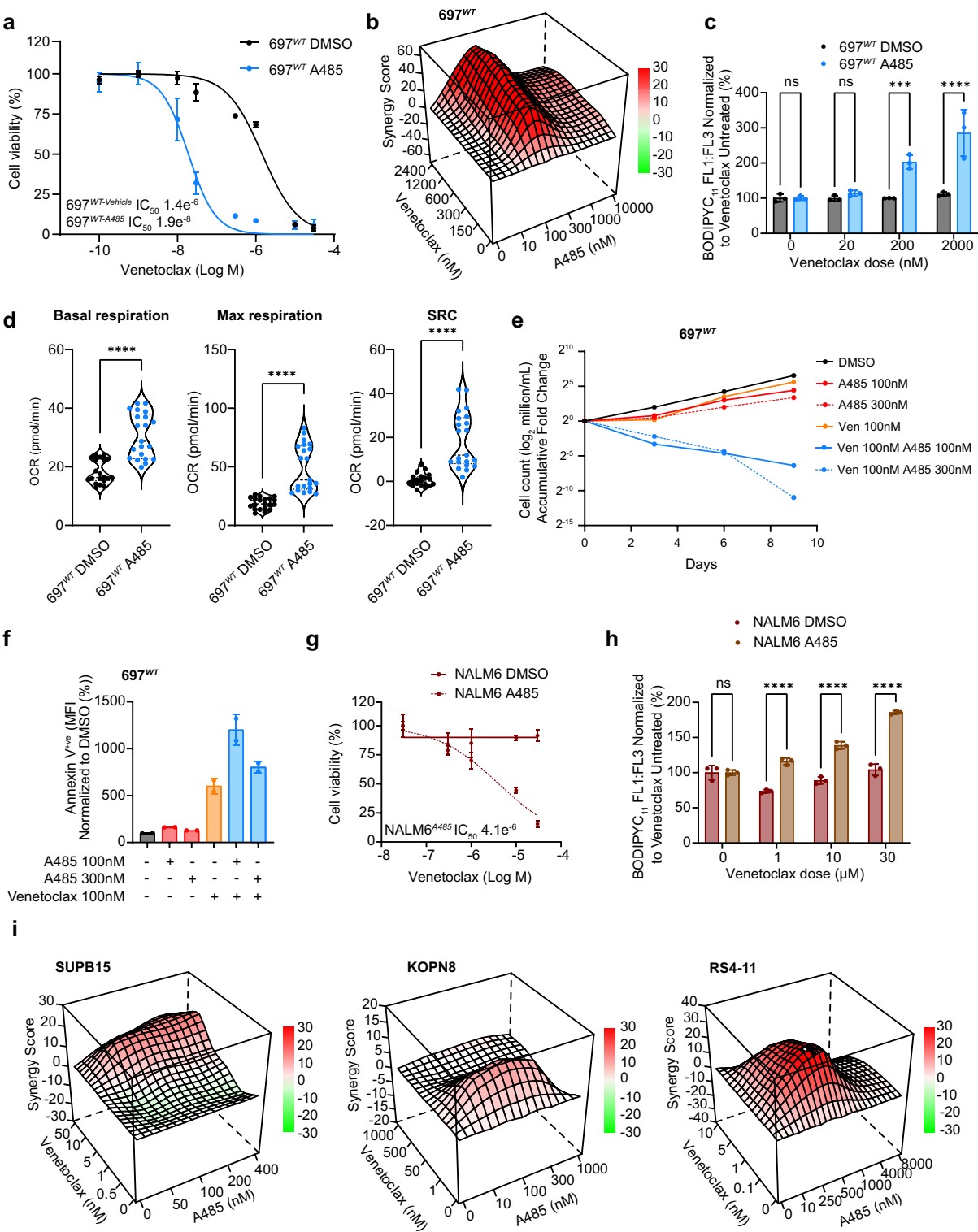

114 d $p = 0.0024$), although the majority of animals still succumbed to disease within the 10-week treatment window. In contrast, 6/7 Combination-treated animals survived until the end of the treatment window (161 vs. 114 d $p = 0.0003$), resulting in a significant OS advantage compared to the Venetoclax only arm ($p = 0.0063$). Histological examination of B-ALL-infiltrated splenic tissue from a separate cohort of mice engrafted with this PDX and that received just two treatment doses showed evidence of 4HNE adducts in the treated mice, consistent with the induction of ferroptosis in vivo (Supplementary Fig. 9f).

Overall, these findings demonstrate that: (i) oral Venetoclax can be highly efficacious in controlling *CREBBP*-mutated B-ALL in pre-clinical in-vivo models; and (ii) pharmacological inhibition of CREBBP is a tolerable and effective means of sensitizing B-ALL to Venetoclax in-vivo.

## Discussion
There is a pressing clinical need to develop novel treatments for patients with high-risk B-ALL, such as those harboring mutations in *CREBBP*. Using a highly-curated panel of clinically-tractable small

**Fig. 8 | Pharmacological inhibition of CREBBP function can sensitize B-ALL cell lines to Venetoclax in-vitro. a** Dose response curve of 697[WT] (black) and 697[WT] pre-treated with 3 days of A485 (pale blue) to Venetoclax in 72 h MTS viability assays. $n = 3$ technical replicates, mean ± SD. **b** Three-dimensional diffusion plot of ZIP synergy score to combined doses of synchronous A485 and Venetoclax (peak ZIP 62.24) in 697[WT] cells. Viability measured by 72 h MTS assay. **c** Summary BODIPYC$_{11}$ staining 697[WT] (black) and 697[WT] treated with A485 (pale blue) cells in response to Venetoclax. FL1:FL3 signal normalized to DMSO vehicle. $n = 3$ independent replicates, each dot represents a single sample, mean ± SD, two-way ANOVA, [***]$P = 0.0006$, [****]$P = 1.2 \times 10^{-6}$. **d** Summary of basal (left) and maximal (middle) mitochondrial oxygen consumption rate (OCR) and spare respiratory capacity (right) measured using Seahorse (Agilent) Mitostress test in 697[WT] cells treated with A485 (pale blue) or DMSO vehicle (black). Each dot represents a single replicate acquired from two separate experiments. Two-tailed unpaired $t$ test [****], Basal respiration $P = 9.2 \times 10^{-8}$, Maximal respiration $P = 5.5 \times 10^{-8}$, SRC $P = 7.3 \times 10^{-7}$. **e** Growth curve of 697[WT] cells treated with Venetoclax 100 nM and/or A485 at two

different doses (100 nM and 300 nM). Mean of duplicate independent replicates presented as log$_2$ accumulative fold change. **f** Annexin-V (APC) externalization assessed by flow cytometry in 697[WT] cells treated with Venetoclax 100 nM and/or A485 at two different doses (100 nM and 300 nM). Duplicate measurements, each dot represents a single independent replicate, MFI ± SD normalized to DMSO control. **g** Dose response curve of NALM6[WT] (solid line) and NALM6[WT] pre-treated with 3 days of A485 (hashed line) to Venetoclax in 72 h MTS viability assays. $n = 3$ technical replicates, mean ± SD. **h** Summary BODIPYC$_{11}$ staining NALM6[WT] (red) and NALM6[WT] treated with A485 (brown) cells in response to Venetoclax. FL1:FL3 signal normalized to DMSO vehicle. $n = 3$ independent replicates, each dot represents a single sample, mean ± SD, two-way ANOVA, [****]$1\,\mu M\ P = 4.9 \times 10^{-7}$, $10\,\mu M\ P = 5.6 \times 10^{-8}$, $30\,\mu M\ P = 4 \times 10^{-11}$. **i** Three-dimensional diffusion plot of ZIP synergy score to combined doses of A485 and Venetoclax. Viability measured by 72 h MTS assays in Venetoclax-sensitive cell lines: SUPB15 (*BCR::ABL1*) (peak ZIP 13.86), KOPN8 (*MLL::ENL*) (peak ZIP 12.66) and RS4-11 (*MLL::AF4*) (peak ZIP 27.08). Source data are provided as a Source Data file.

molecules in an isogenic human cell line model, we demonstrate that *CREBBP* mutation sensitizes cells to Dexamethasone, inhibitors of residual CREBBP/EP300 function and most potently BCL2 inhibition. *CREBBP* LOF profoundly alters metabolism and upon BCL2 inhibition results in ferroptotic cell death, which can be phenocopied by pharmacological CREBBP inhibition in genetically diverse B-ALL cell lines, providing a readily translatable synergistic drug combination for B-ALL more broadly.

Our model shows a number of significant similarities to previously published work. The global transcriptional profile of our *CREBBP*-mutated cells strongly overlaps with those seen in murine *Crebbp*[-/-] germinal-center B-cells[27]. Other models of *CREBBP*-mutated B-ALL and B-cell lymphoma have implicated changes in cell cycle, metabolism, DNA damage response and apoptosis[3,10,20]. We also note that hypodiploid B-ALL, a genetic subtype highly associated with *CREBBP* mutations, has been linked with Venetoclax sensitivity[22]. Moreover, preliminary results from an early phase clinical trial of Venetoclax in relapsed pediatric malignancies (NCT03236857) have highlighted responses in high-risk patients, including the small number of *CREBBP*-mutated B-ALL patients enrolled, providing supportive evidence that our results will be clinically translatable[43].

Our *CREBBP*-mutated model did not show significant differential sensitivity to cytotoxic chemotherapy in the ex-vivo setting. Our cell lines also showed a small but significant sensitization to Dexamethasone, suggesting that *CREBBP* LOF does not significantly alter responses to glucocorticoids[4,11,12]. Previous reports have shown variable association of *CREBBP* LOF with chemoresistance in-vitro[10,12,20] and overall, our findings add to a growing body of evidence that CREBBP LOF does not in itself provoke significant cell-intrinsic chemoresistance.

We show that BCL2 inhibition results in ferroptotic cell death in the context of *CREBBP* LOF. Unlike apoptosis, ferroptosis is not mediated by a defined biochemical pathway, rather is the output of a combination of underlying metabolic state, ROS scavenging capacity and lipid composition[44]. B-cell progenitors are exquisitely sensitive to redox balance[45] and recent genetic perturbation data has highlighted an underlying propensity for ferroptotic cell death in B-ALL[33]. Reflecting its pleiotropic role in biology[46], how *CREBBP* loss alters this balance is likely to be multifactorial, including by direct transcriptional changes, as well as post translational regulation of proteins affecting metabolism and redox balance. Our functional experiments show significant increases in both glycolytic and mitochondrial metabolism upon CREBBP LOF, associated with enhanced sensitivity to inhibition of ROS scavenging pathways. Thus, in contrast with studies in AML, we show an association between enhanced metabolic state and BCL2 dependence, likely relating to a B-cell-state specific susceptibility to oxidative stress[25,33]. Surprisingly, our proteomic analysis identified up-regulation of multiple antioxidant defense pathways, which we

hypothesize to be a compensatory adaptation to the enhanced sensitivity to redox stress induced by CREBBP loss of function. Consistent with this, lipidomic analysis of our 697[WT] and 697[KI] cell lines confirmed a significant increase in the number of ferroptosis-predisposing PUFAs in structural membrane lipids upon *CREBBP* mutation. Intriguingly, we also identify a marked lack of ether-linked lipid species, consistent with the transcriptionally-mediated down-regulation of *AGPS* and protein level reduction of GNPAT and ACSL4, accompanied by increased levels of FAR1, which is post-translationally negatively regulated by plasmalogens[47]. Concordantly, acquired resistance to Venetoclax is associated with reduced rates of oxidative phosphorylation and transcriptional de-repression of *AGPS*, which would be expected to directly affect plasmalogen biosynthesis[48]. Using *CREBBP* expression level as a surrogate for its activity, we also identified highly significant correlations with ROS scavenging pathways and peroxisomal lipid regulators in patients with low *CREBBP* expression. A number of recent papers have linked the enzymatic regulators of plasmalogens to ferroptosis, with variable effects depending on cell type and species studied[36,37,39]. Our unexpected association of ether-linked lipid dysregulation with ferroptosis susceptibility in B-ALL provides a potential new avenue for further therapeutic investigation.

Lastly, we demonstrate that small molecule inhibition of CREBBP can sensitize genetically-diverse B-ALL cell lines to Venetoclax. This potentially widens the paradigm of our combination to multiple ALL genotypes. Furthermore, the clinically-actionable combination of Venetoclax and Inobrodib was tolerable and highly efficacious in cell lines and PDX in-vivo models. We are aware of preliminary reports of this combination being trialed in other related hematological malignancies[49,50] and propose this as a rational drug combination across a wide range of B-ALL patients.

## Methods

### Cell lines
The B-ALL cell lines 697 (ACC 42), REH (ACC 22) and NALM6 (ACC 128) were purchased from DSMZ. RS411, KOPN8 and SUPB15 were provided by Prof. Owen Williams (UCL, UK). 697, REH, KOPN8, RS4-11 were cultured at 37 °C with 5% CO2 in IMDM (ThermoFisher, 12440053) supplemented with 10% v/v heat-inactivated fetal bovine serum (HI-FBS) (Sigma Aldrich, F7524), Penicillin-Streptomycin 100U/ml (Sigma Aldrich, P0781) and 1% v/v L-glutamine (Sigma Aldrich, G7513). NALM6 and SUPB15 were cultured in IMDM 20% serum. HEK-293T cells were maintained in DMEM (Sigma D0819), 10% FBS, L-glutamine, and penicillin/streptomycin. Cell lines were tested for mycoplasma by PCR (CSCI Core Facility) and STR genotyped using Promega Powerplex 16-HS kit (performed by Genetica Lab-Corp). Cells were routinely passaged to $1 \times 10^6$/ml every 2–4 days. Venetoclax-resistant 697[WT] and 697[KI] cells were generated over 10 weeks by exposure to increasing concentrations of Venetoclax

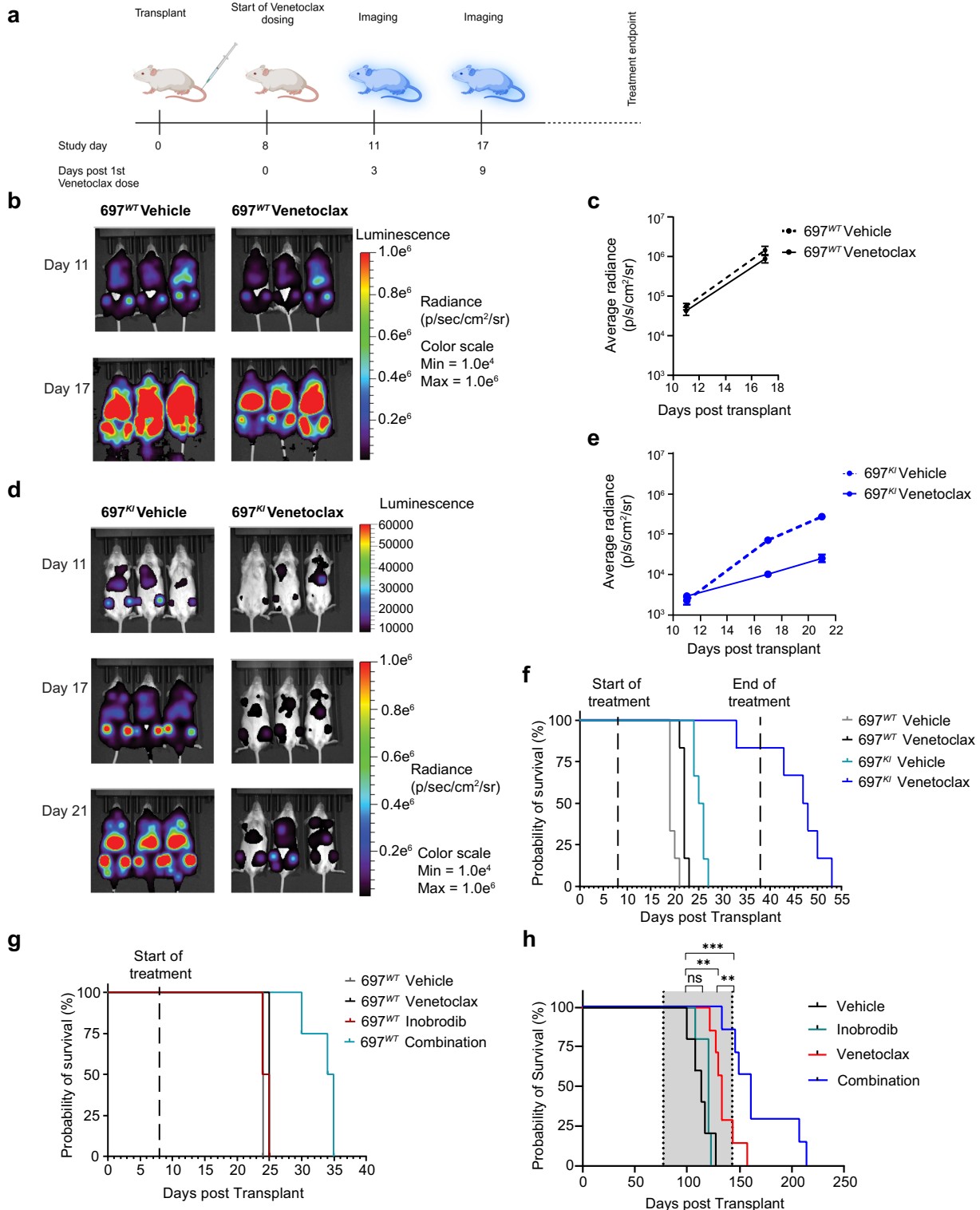

(Selleckchem, S8048) until stable growth in 2000nM Venetoclax. Resistant cells were washed to remove Venetoclax at least 24 h prior to functional experiments.

## Generation of *CREBBP* Mutant Lines

697 and REH cells were lentivirally-transfected with Cas9, single cell cloned and tested for high Cas9 activity by GFP/BFP reporter assay. *CREBBP*$^{R1446C}$ gene-editing reagents were introduced by nucleofection (Nucleofector II, Kit R, Lonza) consisting of a gRNA/Cas9 RNP complex (IDT 1072532, 1081060: target sequence CATGGTAAACGGCTGTG

CGG), two AltR ssDNA Ultramers silently mutating the PAM sequence with or without insertion of the R1446C mutation (50:50 mix: T*T*CACATACTCTAAATATCCAATAAGGATCTCATGGTAAACGGCTGT GCAGAGACAACGTGGCCGGAAGAAATGAATACTATCCAGATAAGAAA TGTA*C*A; T*T*CACATACTCTAAATATCCAATAAGGATCTCATGGTAA ACGGCTGTGCGGAGACAACGTGGCCGGAAGAAATGAATACTATCCA-GATAAGAAATGTA*C*A), 3 mM homology directed repair enhancer (IDT, 1081072) and IDT nucleofection enhancer oligonucleotide (IDT 1075915) according to manufacturer's protocol. At 48 h ATTO550$^{+ve}$ single cells were FACS sorted into 96-well plates (BD FACSAria Fusion

**Fig. 9 | Genetic or pharmacological inhibition of CREBBP sensitizes B-ALL to Venetoclax in-vivo. a** Schema of 697$^{WT}$ vs. 697$^{KI}$ in vivo drug dosing protocol. NSG mice were intravenously injected with 697$^{WT}$ vs. 697$^{KI}$ luciferase-expressing cell lines. At 8 days post injection animals were administered Venetoclax (100 mg/kg) or vehicle control by oral gavage. Mice were imaged by BLI as indicated. $n = 6$ per group. Created in BioRender. Huntly, B. (2025) https://BioRender.com/7ikpp8s. **b** Representative BLI images of mice engrafted with 697$^{WT}$ cells at days 11 and 17 of treatment. $n = 6$ mice per group. **c** Average BLI radiance of mice engrafted with 697$^{WT}$ cells at days 11 and 17 of treatment (log p/s/cm²/sr). Mean ± SEM. $n = 6$ mice per group. **d** Representative BLI images of mice engrafted with 697$^{KI}$ cells at days 11, 17 and 21 of treatment. $n = 6$ mice per group. **e** Average BLI radiance of mice engrafted with 697$^{KI}$ cells at days 11, 17 and 21 of treatment (log p/s/cm²/sr). Mean ± SEM. Vehicle ($n = 5$) and Venetoclax treated ($n = 6$). **f** Overall survival of NSG mice engrafted with 697$^{WT}$ vs. 697$^{KI}$ B-ALL cells treated with oral Venetoclax

(100 mg/kg) or vehicle control. 30 day maximal treatment window is indicated. $n = 6$ per group. Significance calculated by Mantel Cox log-rank test. **g** Overall survival of NSG mice engrafted with 697$^{WT}$ B-ALL cells treated with oral Inobrodib (20 mg/kg) or a combination of Inobrodib (20 mg/kg) plus Venetoclax (100 mg/kg) ($n = 4$ per group). Engraftment control groups were treated with Venetoclax only ($n = 2$) or vehicle control ($n = 1$). Treatment began at day 8 post-transplant. Significance calculated by Mantel Cox log-rank test. **h** Overall survival of NSG mice engrafted with *CREBBP*-mutated B-ALL high-risk PDX cells orally treated with Vehicle control (black, $n = 5$), Inobrodib (20 mg/kg) (green, $n = 5$), Venetoclax (100 mg/kg) (red, $n = 7$) or a combination of Inobrodib (blue, 20 mg/kg) plus Venetoclax (100 mg/kg) ($n = 7$). Treatment began at day 77 post-transplant for 3 consecutive days per week for up to 10 weeks. Significance calculated by Mantel Cox log-rank test, **$P = 0.0063$; **$P = 0.0024$;***$P = 0.0003$. Source data are provided as a Source Data file.

cell sorter) and, upon growth, colonies screened for successful editing by HpyCH4V amplicon digestion (F:GAGCACCTGGAAAGAGGAGC; R:CCCACAGGCGTGTGTACATT). Clones were validated by Sanger sequencing of the bulk screening amplicon and 10xTOPO-TA-cloned PCR products (ThermoFisher, K4575J10).

## MTS
All drugs were purchased from Selleckchem, except A1155463 (Cayman Chemicals), and reconstituted to 10 mM in DMSO. 72-h viability assays were performed using CellTiter 96® AQ$_{ueous}$ One Solution Cell Proliferation Assay (Promega, G3580) following manufacturer recommendations and 490 nM absorbance measured using a CLARIOstar plate reader (BMG Labtech). Drug synergy scores were calculated using SynergyFinder web tool (https://synergyfinder.fimm.fi)[51]. We defined a peak ZIP threshold of +10 for synergism.

## Mitochondrial depolarization using JC1
Mitochondrial depolarization was assessed on cells exposed to Venetoclax for 24 h using the MitoProbe™ JC-1 assay kit (ThermoFisher, M34152) following manufacturer recommendations. Cells were analysed by flow cytometry (BD Fortessa) for 488nm-excited 530/30 filter signal. Viable cells were gated by DAPI (BD, 564907) exclusion. During flow cytometry cells were routinely handled in DPBS (Gibco, 14190-094) 2% HI-FBS and 2 mM EDTA (Invitrogen, 15575038).

## Annexin-V staining by flow cytometry
Annexin-V externalization was assessed using anti-AnnexinV-APC kit from (eBioscience™ 88-8007-74) following manufacturer protocol. Briefly, after incubation with drug/vehicle, cells were washed in DPBS, resuspended in Annexin binding buffer and incubated with 1/40 Annexin-V-APC dilution for 15 min at room temperature (RT) in the dark. Cell pellet was resuspended in 7-AAD (BD, 559925) diluted 1/100 in 1x binding buffer and analyzed by flow cytometry.

## Western blot
Protein lysates were obtained from $1 \times 10^6$ cells after resuspending the pelleted cells in 50 µl boiling lysis buffer (31 mg/ml DTT, 10% SDS, 10% Glycerol, 1% Bromophenol blue and 12.5% 1MTris-HCL) or RIPA buffer (50 mM Tris pH8, 1 mM EDTA 0.5 M, 150 mM NaCl 5 M, 1% NP40(10%), 0.1% SDS(20%) and 0.5% DOC). RIPA buffer lysates were resuspended in boiling loading control with DTT (Cell Signaling Technology, 7722). Cell lysates were separated by SDS-PAGE electrophoresis (NuPAGE 3–8%TRIS acetate protein gels (Invitrogen) or 4-20% Mini-PROTEAN TGX protein gel (Bio-Rad)). Precision-plus protein dual color ladder (Bio-Rad, #1610374) or HiMark™ Pre-stained Protein Standard (Invitrogen, LC5699) were used as a size marker. Proteins were transferred to a methanol pre-activated membrane (Merck, IPFL00010) and then blocked with 5% BSA (ThermoFisher, BP1600) or powdered milk in TBS-0.05% Tween20 (TBST) before proceeding to antibody staining. Primary and secondary antibodies were diluted in 5% BSA or milk in

TBST. Primary antibodies were incubated overnight at 4°C or 1 h at RT; secondary antibodies were incubated for 1 h at RT. Washes were performed with TBST. Membrane fluorescence was acquired using Odyssey imager (LI-COR).

Antibodies used: Caspase 3/PARP (Abcam, Ab136812, 1/250), BCL2 (E17) (Abcam, ab32124, Lot: GR3232704-4, 1/1000), β-Tubulin (Sigma-Aldrich, T8328, 1/2000), 4HNE (Abcam, ab46545, Lot: 1073022-2, 1/1000), GPX4 (Abcam, ab41787, Lot: 1047829-4, 1/1000), CREBBP (A22) (Santa Cruz biotechnology, SC369, Lot: G1613, 1/500) and Vinculin (H-10) (Santa Cruz biotechnology, sc25336, Lot: I2718, 1/250).

## Lentiviral production
Lentiviral vectors were produced in HEK-293T cells by co-transfection of the previously sequenced vector constructs with psPAX and pMDG.2 packaging plasmids using Trans-IT LT-1 (Mirus, MIR2700). HEK293T cells were incubated overnight at 37 °C and the next day medium was changed for IMDM 10% HI-FBS. Viral particles were harvested at 48 and 72 h by centrifugation and 45µm filtration. 250 µl of viral particles were used to transfect 1x10⁶ B-ALL cells by spinoculation (900 g for 2 h at 32 °C) in the presence of 10 µg/ml Polybrene (sc-134220). Subsequently, the cultures were diluted in fresh media and washed three times the following morning.

Transfection efficiencies were assessed by flow cytometric reporters and transfected cells selected using either 0.5 µg/ml of Puromycin (Gibco) for three days or FACS cell sorting as appropriate to achieve >90% transfection.

## shRNA Knockdown of *BCL2*
Doxycycline-inducible reported shRNA knockdown constructs (pLT3GEPIR_GFP/Cherry_Stuffer) were provided by Dr Thomas Mercher (originally from Prof Johannes Zuber), *EcoRI/XhoI* double digested and the vector backbone purified by gel purification as previously described[26]. *BCL2* targeting shRNAs were synthesized as Ultramers (IDT) based on the following recommended sequences (BCL2_1: TGCTGTTGACAGTGAGCGCCCGGGAGATAGTGATGAAGTATAGTGAAGCCACAGATGTATACTTCATCACTATCTCCCGGTTGCCTACTGCCTCGGA; BCL2: TGCTGTTGACAGTGAGCGCGAGGATCATGCTGTACTTAAATAGTGAAGCCACAGATGTATTTAAGTACAGCATGATCCTCTTGCCTACTGCCTCGGA; Renilla: TGCTGTTGACAGTGAGCGCAGGAATTATAATGCTTATCTATAGTGAAGCCACAGATGTATAGATAAGCATTATAATTCCTATGCCTACTGCCTCGGA)[26]. Complementary oligonucleotides were resuspended to 500 ng/µl, boiled and hybridized, ligated using T4 ligase (NEB) (5 ng insert:50 ng vector) and transformed into chemically-competent *E. coli*. Bacterial colonies were screened by PCR (F:CTCGACTAGGGATAACAGGG R:CAAAGAGATAGCAAGGTATTCAG) and successful plasmid clones miniprepped (Qiagen) and sequence verified by Sanger sequencing. Cells were selected with puromycin 0.5 µg/ml. Cells were incubated with 500 ng/ml of doxycycline and mCherry positive cells FACS sorted to confirm knockdown by qPCR and western blot.

Competitive co-culture assays were performed by mixing an equal number of *renilla* shRNA-GFP and one of two *BCL2* shRNA-mCherry transfected 697$^{WT}$ or 697$^{KI}$ cells in triplicate. *BCL2* knockdown was induced with doxycycline and aliquots of cell culture analyzed daily by flow cytometry. Results are presented as the ratio of 697$^{KI}$ Cherry:GFP events normalized to the mean ratio of 697$^{WT}$ Cherry:GFP events.

## Gene expression by reverse transcriptase quantitative PCR (RT-qPCR)

Total RNA was extracted using the RNeasy kit (Qiagen) and 1 µg was used to synthesize cDNA using TaqMan Reverse transcription (Applied Biosystems, N8080234) following manufacturer's instructions. RT-qPCR was carried out with SYBR Green PCR master mix (Agilent, 600882) on cDNA (diluted 1:5 in water). The expression level of RNA was calculated using the standard curve method, normalized to the expression of *GAPDH*. Primer pairs: *BCL2*F:ATTGGGAAGTTTCAAAT-CAGC; *BCL2*R:TGCATTCTTGGACGAGGG; *GAPDH*F:CCACATCGCTCA-GACACCAT; *GAPDH*R:CCAGGCGCCCAATACG.

## Cell cycle analysis by FUCCI reporter

CSII-EF-MCS vectors encoding mCherry-hCdt1(30/120) and mVenus-hGeminin(1/110) were obtained from Atsushi Miyawaki, RIKEN BioResource Research Center[52,53]. Lentiviral vectors were made and dual transfected into 697 isogenic cell lines as above. mCherry/mVenus dual-positive cells were enriched by FACS cell sorting (BD Influx) to generate high-expressing stable cell lines and analyzed by flow cytometry.

## Seahorse metabolic assays

Oxphos and glycolysis assays were performed using the Seahorse XFe96 analyser (Agilent) according to manufacturer's instructions. Briefly, cells were analysed at 48-72 h post passage and in drug concentrations/DMSO vehicle as stated in figure legends. For glycostress experiments, cells were harvested and counted in Seahorse XF base medium, pH7.4 supplemented with 2mM L-Glutamine (Sigma Aldrich, G7513). The final concentration of the injected drugs in the glycostress test is 10 mM glucose (Sigma, G7021), 1 µM Oligomycin A and 50 mM of 2-deoxyglucose (2-DG) (Sigma, D3179). For mitostress tests, Seahorse XF base media was supplemented with 2mM L-Glutamine, 10 mM Glucose and 1 mM Sodium pyruvate (Sigma, S8636). The final concentration of the injected drugs was 1 µM Oligomycin, 1 µM FCCP supplemented with 1 µM Sodium pyruvate and 1 µM Antimycin. For experiments using Venetoclax or A485 pretreatment drug exposure continued in the Seahorse media. Cells were plated at a density of 70000 cells/well in XFp tissue culture plates previously coated with CellTak (Corning, 354240) and incubated for 45 mins at 37 °C in a CO2-free incubator prior to analysis. Seahorse experiments were performed with multiple biological replicates on at least two separate occasions to account for technical failures or gross outliers, which were excluded based on the independent judgment of two researchers.

## Lipid peroxidation assays

Lipid peroxidation was assessed using the lipid peroxidation sensor BODIPYC$_{11}$-581/591 (ThermoFisher, D3861) according to manufacturer's instructions. Briefly, cells were washed in DPBS and stained with 4 µM BODIPY dye for 30 min at 37 °C with gentle shaking in the dark. After washing with DPBS, cells were analyzed by flow cytometry. Replicate data is presented as the ratio of 488nm-excited 530/30 (FL1) to 610/20 (FL3) filtered signal.

## Proteomics

Pellets were lysed in a buffer containing 100 mM triethylammonium bicarbonate (TEAB) (Sigma, #T4708), 1% sodium deoxycholate (SDC), 10% isopropanol, 50 mM NaCl, protease and phosphatase inhibitors

(Halt, #78441) and universal nuclease (Pierce, #88700), and the protein concentration was measured with Bradford assay (Bio-Rad, Quick Start™). 100 µg total protein per sample were reduced and alkylated simultaneously by adding tris-2-carboxyethyl phosphine (TCEP, Sigma) at a final concentration of 5 mM, and freshly prepared iodoacetamide at a final concentration of 10 mM followed by 1 h incubation at RT in the dark. Then, the samples were digested overnight at 37 °C with trypsin (Pierce #90058) and peptides were labeled with the TMTpro-16plex reagents according to manufacturer's instructions (Thermo Scientific). The TMT mixture was fractionated on a Dionex UltiMate 3000 system at high pH using the X-Bridge C18 column (3.5µm 2.1x150mm, Waters). Twenty-two https://www.thermofisher.com/order/catalog/product/A44520 fractions were collected and 10% was transferred on different tubes for the TMT whole-proteome analysis.

Whole proteome analyses were performed on a Dionex UltiMate 3000 UHPLC system coupled with the nano-ESI Fusion-Lumos (Thermo Scientific) mass spectrometer. The MS2 scans were performed in the ion trap with collision energy 32%. Peptides were isolated in the quadrupole with isolation window 0.7 Th. The 10 most intense fragments were selected for Synchronous Precursor Selection (SPS) HCD-MS3 analysis with MS2 isolation window 2.0 and HCD collision energy 50%. Raw files were processed with the SequestHT search engine on Proteome Discoverer 2.4 software. The SequestHT node included the following parameters: Precursor Mass Tolerance 20 ppm, Fragment Mass Tolerance 0.5 Da for the CID spectra and 0.02 Da for the HCD spectra, Dynamic Modifications were Oxidation of M (+15.995 Da), Deamidation of N, Q (+0.984 Da) and Static Modifications were TMTpro at any N-Terminus, K (+304.207 Da) and carbamidomethyl at C (+57.021). Only unique peptides filtered for FDR < 1% were used for quantification and the consensus workflow included calculation of TMT signal-to-noise.

Data processing, normalization, and statistical analysis of peptide intensities were performed using the qPLEXanalyzer package from Bioconductor (cite: https://www.nature.com/articles/s41467-018-04619-5). Peptide intensities were normalized using median scaling. Protein-level quantification was then determined by summing the normalized peptide intensities. A statistical analysis of differentially regulated proteins was subsequently conducted using the two-sided limma statistical test. To control for the false discovery rate (FDR), p values were adjusted using the Benjamini-Hochberg method for multiple testing correction.

## Lipidomics

Lipids were extracted using the protein-precipitation liquid extraction protocol has been described previously[54]. At each pellet of cells, 650 µL of Chloroform, 100 µL of Lipids internal standard (5 µM in methanol) was added. Then 250 µL of methanol was added to each sample and finally 400 µL of acetone was added. The samples were vortexed for 60 s and then centrifuged for 10 min. The single-phase supernatant was transferred in other multiwall plate. The extract (-1.4 mL) of each sample was dried using gentle flow of N$_2$ at 60 °C. The samples were reconstituted in 100 µL of 2:1:1 (propan-2-ol, acetonitrile and water, respectively) then thoroughly vortexed. The samples were analysed using LC-MS, with reversed phase chromatography and high-resolution mass spectrometry in both positive and negative mode electrospray ionization. Full chromatographic separation of the lipids was performed using a Waters HPLC system. 10 µL of sample extraction was injected onto a Waters Acquity UPLC CSH C18 column (1.7 µm, I.D. 2.1 mm × 50 mm), maintained at 55 °C. Two mobile phase have been used: mobile phase A consists of acetonitrile, and water 6:4 with 10 mM ammonium formate; mobile phase B consists of propan-2-ol, and acetonitrile 9:1, with 10 mM ammonium formate. The flow was maintained at 500 µL per minute through the gradient[54].

The raw data was processed using MS-DIAL software (version 4.9). This process included peak selection, alignment and data annotation[55]

(see Supplementary Data Table 3 for parameters). The database of the lipid was validated against Lipid Maps (http://www.lipidmaps.org/tools/ms). The annotation of lipid molecular species was performed using the accurate mass data acquired with mass spectrometry (MS) and specific retention time acquired from the liquid chromatography. Lipid semi-quantification was performed using the deuterated internal standard for each lipid class and was based on the known concentrations of the standards added to the samples (Supplementary Data Table 3). At the end of this analytical step, a set of matrices was produced containing the expressive concentrations in nmol/1 million cells of the lipid molecular species previously annotated. For the lipidomic data analysis and data matrices preparation LipidOne 2.0 web-platform was utilized[56,57].

## RNA sequencing library preparation

$697^{WT}$ and $697^{KI}$ treated with 20 nM Venetoclax or DMSO vehicle for 24 h (Fig. 4) or $697^{WT}$, $697^{KI}$, $697^{WT-VenR}$ or $697^{KI-VenR}$ cells removed from Venetoclax for 24 h (Fig. 5) were harvested, washed and RNA was extracted from $4 \times 10^6$ cells using the RNeasy kit (Qiagen) (performed in biological triplicate). Libraries were prepared using NEBNext Poly(A) mRNA magnetic Isolation module (NEB, E7490) starting with 1 μg DNA-free RNA, as per manufacturer's instructions. RNA fragmentation, double-strand cDNA synthesis, end repair, adapter ligation and PCR amplification (9 cycles) was performed using NEBNext Ultra II directional RNA library prep kit for Illumina protocol (NEB, E7760) according to manufacturer's protocol. Library quality and molarity was measured by Qubit (ThermoFisher) and TapeStation (Agilent). Samples were 50-bp paired-end sequenced on the NovaSeq (Illumina) instrument (CRUK CI Genomics Core Facility).

## RNAseq analysis

The quality of the paired-end RNA-seq reads were assessed using FastQC (https://www.bioinformatics.babraham.ac.uk/projects/fastqc/). All RNA-seq libraries were found to be above the minimum quality thresholds across quality control metrics. Adapter sequences were trimmed using TrimGalore package (https://github.com/FelixKrueger/TrimGalore), and were next mapped against the human genome version 38 (Hg38) using STAR Version=2.7.10a[58]. Only uniquely-mapping high confidence reads (flags NH:i:1 and MAPQ = 255) were retrieved and used for transcriptome quantification with featureCounts version 2.0.1[59]. Genes with a minimum of 10 reads or more in a minimum of two samples were considered for all downstream analysis. Differential expression analysis was based on either a single factor model ( ~ genotype/treatment/sensitivity) or an interaction model when two factors were considered ( ~ genotype * treatment), and the interaction term was used to determine the difference of the effect of treatment across genotypes. The interaction model was chosen over the additive model for two-factor analysis of differential expression based on the clustering of normalized RNA-seq libraries on a PCA plot and the number of statistically significant differentially-expressed genes across both models using likelihood-ratio test (LRT). Differential expression analysis was conducted using gene level transcriptomic reads using the Bioconductor package DESeq2[60]. KEGG Pathway enrichment, Gene Set Enrichment Analyses (GSEA) and Gene Ontology (GO) Analyses were all performed using the R package 'clusterProfiler' version 4.4.4[61]. A minimum $p$ value (corrected for multiple testing) and FDR of 0.05 were used across all tests to establish statistical significance.

The correlation of *CREBBP* expression in human patients with genes/pathways of interest was based on the RNA-seq cohort of B-ALL patients from the Therapeutically Applicable Research to Generate Effective Treatments (TARGET) Phase II initiative (https://www.cancer.gov/ccg/research/genome-sequencing/target)[38]. Reads-per-kilobase-million (RPKM) normalized reads from 203 patients were obtained from TARGET's Genomic Data Commons (https://portal.gdc.cancer.gov) portal along with associated clinical and sample metadata. The

median average expression of all genes in pathways of interest was compared with *CREBBP* expression for each patient, and the results were collated for correlation across all patients. All $p$ values were corrected for multiple-testing using the Bonferroni formula.

## ChIP-Seq/CUTandRUN

7 million $697^{WT}$ and $697^{KI}$ were cross-linked with 1% formaldehyde for 10 min at RT, reaction that was stopped by Glycine (final concentration 125 mM). Cells were washed twice with ice-cold PBS and finally resuspended in cold PBS with 1x Protease inhibitors (PI) (Roche, 11836170001). After this step all buffers contained PI. Cell pellets were lysed with cell lysis buffer (10 nM Tris-Cl, 10 mM NaCl, 0.2% NP40) for 10 min at 4 °C. After centrifugation at 2400 rpm at 4 °C, nuclei were resuspended in 100 μl of Sonication buffer (0.5% SDS, 20 mM Hepes, 5 mM EDTA) and incubated on ice for 10 min. Chromatin was sheared by sonication (three cycles of 30 s ON and 30 s OFF). Prior to immunoprecipitation, the sonicated chromatin was diluted with 400 μl of dilution buffer (20 mM Tris pH8.1, 2 mM EDTA, 150 mM NaCl, 1% TritonX100 and 0.01% SDS). Immunoprecipitation was performed overnight in 20% Tx-100 using 2.5 μg of antibody (IgG, Proteintech, 30000-0-AP; H3 (1B1B2), Cell Signaling Technology, 14269S and H3K27ac, Active motif, 39133). Immunocomplexes were captured using the protein A (10002D) and protein G (10004D) Dynabeads (Invitrogen). After 4 h, the immunocomplexes were magnetized and repeatedly washed in the following buffers: RIPA buffer (20 mM Hepes, 150 mM NaCl, 1 mM EDTA, 0.1% SDS, 1% Tx-100), RIPA-500 (20 mM Hepes, 500 mM NaCl, 1 mM EDTA, 0.1% SDS, 1% Tx-100, 0.1% DOC) and LiCl wash buffer (10 mM Tris-HCl, 1 mM EDTA, 150 mM LiCl, 0.5% NP40, 0.5% DOC).

To perform reverse crosslinking, ChIP samples were incubated overnight at 65 °C with ChIP elution buffer (10 mM Tris-HCl, 5 mM EDTA, 300 mM NaCl, 0.4% SDS) with 2 μl of Proteinase K (Thermo Scientific EO0491). Samples were cleaned up using a Qiagen PCR Minelute column.

To perform CUTandRUN, 0.3 million cells per sample of $697^{WT}$ and $697^{KI}$ cell lines were harvested and washed in wash buffer (20 mM HEPES, 150 mM NaCl, 0.5 mM Spermidine, 1x cOmplete™ EDTA-free PI). Cells were incubated for 10 min at RT with activated Concavalin A beads (EpiCypher, 21-1401). The suspension was magnetized, supernatant removed and the cells were resuspended in 1 μg of the corresponding antibody (CREBBP, Santa cruz technologies, sc369(A22), lot: G1613; IgG, Epicypher, cat 13-0042) diluted in antibody buffer (wash buffer plus 0.01% digitonin and 2 mM EDTA) on a nutator overnight at 4 °C. After incubation, the suspension was magnezited and beads were washed and finally resuspended with digitonin buffer (wash buffer plus 0.01% digitonin) prior to the 10 min incubation of the nuclei with CUTANA pAG-MNase (EpiCypher, 15-1016). Beads were washed again with digitonin buffer and the targeted chromatin was digested and released with the addition of 100 mM $CaCl_2$ that activates the MNase tethered to chromatin. After 2 h at 4 °C, stop buffer (40 mM NaCl, 20 mM EDTA, 4 mM EGTA, 50 μg/ml RNase A and 50 μg/ml Glycogen) was added. Cleaved DNA was purified using the Monarch PCR DNA clean up kit (NEB).

Library preparation was performed using the NEBNext Ultra II DNA library preparation kit for Illumina (NEB, E7645) according to manufacturer's protocol. Library quality and molarity was measured by Qubit (ThermoFisher) and TapeStation (Agilent). Samples were 50-bp paired-end sequenced on the NovaSeq (Illumina) instrument (CRUK CI Genomics Core Facility).

## ChIP-Seq/CUTandRUN analysis

Sequencing libraries that passed the minimum quality requirements, as determined by the quality metrics of the tool FastQC (https://www.bioinformatics.babraham.ac.uk/projects/fastqc/), were considered for downstream analysis and processed using either the ChIP-seq or

CUTandRUN pipelines. For ChIP-seq libraries, paired end reads were first trimmed for Trueseq adaptors using trimmomatic[62] (version 0.36) with the following parameters (PE -phred33 ILLUMINACLIP:-Truseq3.PE.fa:2:15:4:4:true LEADING:20 TRAILING:20 SLI-DINGWINDOW:4:15 MINLEN:25). Adaptor-trimmed reads were then aligned to the human genome version hg38 using Bowtie 2[63] (version 2.4.5) using the following parameters (--very-sensitive-local --phred33 -I 10 -X 700). Aligned reads in SAM format were converted to BAM format, sorted then indexed using Samtools[64] (version 1.11). Duplicate reads in aligned BAM files were marked using the Picard tool Mark-Duplicates, and the resulting duplicate reads were subsequently removed by Samtools. The aligned, de-duplicated read files were converted to BigWig files for genome browser visualization and analysis of co-occupancy with CREBBP libraries using the bamCoverage function of the deeptools[65] (version 3.5.1), normalizing by counts-per-million (CPM) and a bin size of 10. CUTandRUN libraries were processed using the CUT&RUNTools pipeline (available at: https://github.com/fl-yu/CUT-RUNTools-2.0)[66], using the same human reference genome version (hg38). We ran the pipeline without spike-in controls, or filtering for short fragments ( < 120 bp) to maximize the capture of CREBBP binding.

Genomic occupancy of CREBBP/H3K27ac were established by peak-calling against matched input controls using the PCR-duplicate filtered, aligned paired-end BAM files from the ChIP-seq and the CUT&RUNTools pipeline. MACS2[67] (version 2.2.7.1) was used to call peaks with the callpeak function and the following parameters (-p 0.05 -B --SPMR --keep-dup all). Black-listed regions were subsequently removed from the called peaks using bedops[68] (version 2.4.41) and a list of version-specific blacklisted region list from the human genome. Peaks with a minimum q-value of 0.05 were utilized for all downstream analysis.

CREBBP and H3K27ac were annotated to their associated genes by their proximity to transcription start sites (TSSs) using the R package ChIPseeker[69]. Promoter regions were defined as the ±3 kb window around TSSs, with any region acetylated by H3K27ac outside of that window and within a ± 100 kb window of a TSS as an enhancer element. Co-occupancy of CREBBP and H3K27ac was determined by intersecting the two peak sets using the R package IntersectionSet[70], and the Venn diagrams of those intersection were generated using the R package eulerr.

Profile plots showing genomic signal enrichment of simultaneous CREBBP/H3K27ac occupancy around TSSs of differentially expressed genes from the RNA-seq experiments of $697^{WT}$ vs. $697^{KI}$ were generated using scores calculated by the computeMatrix function of the deeptools package with the parameters (--binSize 10 --skipZeros). Score files inputs (in BigWig format) for this analysis were generated by converting the total BigWig signal files from the ChIP-seq/CUTandRUN pipelines to Wig format using bigWigToWig from the UCSC utilities[71], then converted to BED format using wig2bed from bedops, followed by selecting those regions that were established to be common between $697^{WT}$ vs. $697^{KI}$ or those with co-bound CREBBP-H3K27ac sites in the previous step, and saved as bedgraph files, to be converted next to BigWig again using bedGraphToBigWig from the UCSC utilities.

The integration of CREBBP binding with the differential gene expression data from $697^{WT}$ vs. $697^{KI}$ to deduce co-regulation of target genes was done using the Binding and Expression Target Analysis (BETA) package[28]. For $697^{KI}$, the entire list of expressed genes with the adjusted $p$ values and log2 fold-change of the difference of their expression to their $697^{WT}$ counterparts, along with the list of CREBBP-H3K27ac enhancer-associated peak regions, were used as input for the BETA plus function with the following parameters (-g hg38 --df 0.05 --da 1 --mn 10). The same analysis was repeated for the $697^{WT}$, but with reversing the signs for the log2 fold-change in gene expression values to account for the directionality in differential gene expression.

## In-vivo dosing studies

All experiments were conducted under a UK Home Office project (under the Animals (Scientific Procedures) Act 1986, Amendment Regulations (2012)) and following ethical review by the University of Cambridge Animal Welfare and Ethical Review Body. Primary patient samples for PDX derivation were supplied by the VIVO Biobank (REC reference 23/EM/0130). Parental consent was obtained and no compensation provided.

For the in-vivo dosing experiments in Fig. 9a–f, $0.25 \times 10^6$ luciferase expressing cells (isogenic cell lines $697^{WT}$ & $697^{KI}$) were injected by tail vein injection into sub-lethally (2 Gy) irradiated 13-15 week-old female NSG (NOD.Cg-Prkdcscid Il2rgtm1Wjl/SzJ) mice. Mice received a daily dose of Venetoclax (100 mg/kg)(LC Labs) or vehicle control (15% Kolliphor HS15 (Sigma 42966), 60% Phosal propylene glycol (MP, 151957) and 10% ethanol), by means of Oral Gavage (FTP-20-30, INSTECH) for thirty days or until the experimental endpoints were reached. Disease progression was tracked by IVIS bioluminescence imaging (PerkinElmer); in brief, D-luciferin (Perkin Elmer, 122799) was administered by intraperitoneal (IP) injection (10 µl/g body weight of a 15 mg/ml solution) followed by inhalation anesthesia (isoflurane) and IVIS bioluminescence imaging ( -10 min post-luciferin injection). Tumor burden was quantified using Living Image Software (v4.7.2, PerkinElmer). One mouse engrafted with $697^{KI}$ treated with Vehicle was excluded due to imaging failure.

For the in-vivo dosing experiments in Fig. 9g, $0.25 \times 10^6$ $697^{WT}$ cells were injected by tail vein injection into sub-lethally (2 Gy) irradiated 11-13 week-old old male NSG mice. Mice received a daily dose of Venetoclax (100 mg/kg) and/or Inobrodib (20 mg/kg) or vehicle control (as above), by means of oral gavage until the experimental endpoints were reached.

For the in-vivo dosing experiments in Fig. 9h, the CREBBP-mutated male, pediatric patient presented was genomically and genetically characterized by the VIVO Biobank as having a high hyperdiploid karyotype alongside heterozygous CREBBP p.(Glu1551_Ser1552-insArgArgPro) and NRAS p.(Gly12Cys) mutations. High quality primary diagnostic cells were obtained from the VIVO Biobank and engrafted by tail vein injection into sub-lethally irradiated NSG mice. For the dosing experiments, 1.2 million secondary PDX cells were engrafted into sub-lethally irradiated NSG mice. Recipients were 17-23 week-old old male and female NSG mice. Animals were randomized by sex and weight (Vehicle 2 M/3 F; Inobrodib 2 M/3 F; Venetoclax 2 M/5 F; Combination 3 M/4 F). Dosing was performed as above from day 77, except the schedule was 3 consecutive days per week for a maximum of 10 weeks. For the histology presented in Supplementary Fig. 9f, mice received two daily doses from D107 after transplantation.

Mice were housed in a pathogen-free animal facility and were allowed unrestricted access to food and water, including irradiated complete universal diet for rats, mice and hamsters (SAFE R105-25), supplemented by DietGel 76 A (Clear $H_2O$ 72-07-5022), NutraGel (BioServ NGB) and/or peanut butter provided across all groups during drug dosing. Housing and care of all animals included in this study were according to the UK's Home Office CoP with a 7am-7pm light/dark cycle (with a dawn/dusk cycle incorporated). Ambient temperature and relative humidity parameters were kept within 19–23 °C & 45–55%, respectively. Mice showing any clinical signs or weight loss (more than 20% of its reference weight post irradiation or 15% after administration of therapeutics), or sustained signs of compromised wellbeing such as reduced activity, piloerection and hunching that persisted for more than 6 h (or 2 health checks separated by at least three hours) were humanely sacrificed. One mouse engrafted with $697^{KI}$ treated with Vehicle was excluded from summary BLI measurements due to imaging failure. No other animals were excluded from analysis. The investigator was not blinded to the group allocation, however, the animal technicians who provided the majority of animal care and who decided upon which animals to sacrifice (according to defined endpoints) were blinded to the allocation.

Sample sizes were based on pilot studies showing median survival times for 697 cells. For the Venetoclax dosing studies in Fig. 9a–f, a sample size of 6 was chosen to detect a $5 \pm 3$ day OS benefit in the treatment arm, based on an alpha of 0.05 and a power of 80% and an anticipated engraftment time of 24 days in the control group. For the combination studies in Fig. 9g, minimal control numbers were used for Venetoclax or vehicle dosing in line with three R principles. The number of mice used for treatment of either Inobrodib or Inobrodib with Venetoclax was calculated as four per arm, based on an anticipated survival benefit of $6 \pm 3$ days in the dual treatment arm, based on an alpha of 0.05 and a power of 80% and an anticipated engraftment time of 24 days in the control group. For the combination dosing studies in PDX in Fig. 9h a minimal sample size of 5 was chosen to detect a $35 \pm 20$ day OS benefit in the treatment arms, based on an alpha of 0.05 and a power of 80% and a predicted latency of 120 days. Power calculations were calculated using ClinCalc.com.

## Histology

Formalin fixed embedded splenic tissue was deparaffinized, rehydrated, antigen retrieved in citrate buffer (pH 6.0) and blocked with hydrogen peroxide. Washes were performed with TBS/Tween20 (0.05%). Primary antibodies (4HNE (Abcam, ab46545, Lot: 1073022-2), GPX4 (Abcam, ab41787, Lot: 1047829-4)) were diluted 1:200 in milk and detected by HRP-conjugate. Nuclei were stained with Harris's Haematoxylin counterstain. Images were obtained on a Zeiss Axioimager M2.

## Statistics

Flow cytometric data was analyzed using FlowJo (v10). Statistical analysis was performed in GraphPad Prism-v9 and Microsoft Excel. Seahorse data was processed on Wave software (2.6.3). Significance tests are stated in figure legends

## Reporting summary

Further information on research design is available in the Nature Portfolio Reporting Summary linked to this article.

## Data availability

Raw sequencing data for RNAseq, ChIPseq and CUR&RUN are deposited in GEO Archive under accession numbers GSE248265, GSE289011 and GSE289012 respectively [https://www.ncbi.nlm.nih.gov/geo/]. The total proteomics data generated in this study have been deposited in the ProteomeXchange database under accession code PXD061362 [https://www.ebi.ac.uk/pride/archive?sortDirection=DESC&page=0&pageSize=20]. The total lipidomics data generated in this study have been deposited in the MassIVE database under accession code MSV000097461 [https://massive.ucsd.edu/ProteoSAFe/dataset.jsp?task=6746d9c3cb2d4c4e957cde41bd03aa08]. Further information and requests for resources and reagents should be directed to and will be fulfilled by the corresponding author. Source data are provided with this paper.

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

## Acknowledgements

S.E.R. is supported by a Clinician Scientist Fellowship from Cancer Research UK (C67279/A27957), a Leukemia UK John Goldman Fellow-ship (2022/JGF/004), a European Hematology Association Physician Scientist Research Grant and funds from the Isaac Newton Trust. We would like to thank funders of the Huntly laboratory including Cancer

Research UK (C18680/A25508, C355/A26819 and DRCRPG-Nov22/100014), the European Research Council (647685), MRC (MR-R009708-1 and MR-X008371) and the Kay Kendall Leukemia Fund (KKL1243 and KKL1440). B.J.P.H. is part of a Strategic Center of Research (7035-24) supported by The Leukemia & Lymphoma Society and Blood Cancer UK. This research was supported by the NIHR Cambridge Biomedical Research Center (BRC-1215-20014) and the Cancer Research UK Cambridge Center (Cancer Research UK Major Center Award C9685/A25117), and was funded in part by the Wellcome Trust, who supported the Cambridge Stem Cell Institute (203151/Z/16/Z and 226795/Z/22/Z) and Cambridge Institute for Medical Research (100140/Z/12/Z). N.N. was funded by a Kay Kendall Leukemia Fund Junior Fellowship (KKL1440). J.B. was funded by the Wellcome Trust 4-Year MRes+PhD Program in Stem Cell Biology and Medicine (218481/Z/19/Z). T.B. was funded by a Clinical Research Training Fellowship from the Cancer Research UK Cambridge Center Clinical Academic Training Program (SEBSTF-2021\100001). M.R.M. is funded by a GOSHCC professorship. K.F. is jointly funded by CRUK and Children with Cancer charity (grant number DRCPGM\100066). Work in the MPM lab was supported by the Medical Research Council UK (MC_UU_00028/4). The views expressed are those of the authors and not necessarily those of the NIHR or the Department of Health and Social Care. For the purpose of Open Access, the author has applied for a CC BY public copyright licence to any Author Accepted Manuscript version arising from this submission. We acknowledge support from the Cancer Research UK Cambridge Center Genomics and Proteomics Facilities, the Cambridge Institute for Medical Research Flow Cytometry Facility, the NIHR Cambridge BRC Cell Phenotyping Hub and the CSCI histology and imaging facilities. We thank Thomas Mercher and Johannes Zuber for use of the pLT3GEPIR inducible shRNA system. We thank Atsushi Miyawaki at the RIKEN BioResource Research Center for the use of the FUCCI cell cycle reporter system. We thank Anthony Moorman for his assistance in identification of *CREBBP*-mutated patient samples. Patient samples used in this study were provided by VIVO Biobank, supported by Cancer Research UK & Blood Cancer UK (Grant no. CRCPSC-Dec21\100003).

## Author contributions

A.G.-G., J.E.D., D.M.A.A., G.G., R.A., E.M., J.B., N.S., C.K.L., N.N., T.B., S.A.S., K.F., H.B.R.A., B.J., A.K., S.J.H. and S.E.R. performed the experiments and analyzed the data. A.G.-G., G.G., D.O.C., M.R.M., H.B.R.A., A.K., M.P.M., S.J.H., B.J.P.H. and S.E.R. wrote, reviewed and edited the manuscript. B.J.H.P. and S.E.R. conceptualized the study, acquired funding and supervised the project.

## Competing interests

N.N. is a former employee of the Walter and Eliza Hall Institute which receives milestone and royalty payments related to Venetoclax. N.N. received payments from WEHI related to Venetoclax. The combination of BCL2 and CREBBP/EP300 inhibitors in ALL is protected under Cambridge Enterprise patent PCT/GB2024/052398 (S.E.R. and B.P.J.H. listed inventors). The remaining authors declare no competing interests.
