## [Transparent Peer Review file · Nature Communications]

CREBBP inactivation sensitizes B cell Acute Lymphoblastic Leukemia to Ferroptotic Cell Death upon BCL2 Inhibition

Corresponding Author: Dr Simon Richardson

Version 0:

Reviewer comments:

Reviewer #1

(Remarks to the Author)

The authors have made a lot of effort to improve this manuscript which is quite commendable and contributes a lot to scientific rigor. They have addressed all of my previous concerns appropriately.

Reviewer #2

(Remarks to the Author)

My points have been addressed sufficiently or rebutted elegantly by the authors. Together with extensive revisions to address the points by the three reviewers, this manuscript now convincing and interesting for readers in cancer biology and all fields in biology that study non-apoptotic cell death. It convey interesting observations with respect to complementarity of CREBBP and EP300. I have no further requests.

Reviewer #3

(Remarks to the Author)

The authors have made a concerted effort to answer the reviewers comments and have performed additional informative experiments that have improved the study. They have also revised the text to reflect the comments and new data. With a few minor changes (see below), I believe the manuscript is now ready for publication in Nature Communications. Congratulations to the authors on a robust and impactful study!

Minor corrections:

In the text (page 6) when discussing Liproxstatin it says that "Here, it and almost completely rescued viability in 697KI cells exposed to low-dose Venetoclax", but in reality, the viability goes from approx. 20 to 40%. The authors should adapt this sentence to say that the rescue is statistically significant but not a complete rescue.

In the text (page 7) it says "697KI cells had markedly higher levels of pro-ferroptotic polyunsaturated fatty acids (PUFAs)", this text should be altered because PUFAs are not pro-ferroptotic, they are more vulnerable to ROS damage and lipid peroxidation, which leads to ferroptosis, so the text should reflect this.

Figure 4i is independent data from the other graphs in the row and needs a key - it doesn't currently have one.

Response to Referees

Re: CREBBP inactivation sensitizes B cell Acute Lymphoblastic Leukemia to Ferroptotic Cell Death upon BCL2 Inhibition (NCOMMS-25-09054-T)

We thank all the reviewers for their positive comments to our revisions. A point by point response to the comments from Reviewer 3 are listed below.

REVIEWERS' COMMENTS

Reviewer #1 (Remarks to the Author):

The authors have made a lot of effort to improve this manuscript which is quite commendable and contributes a lot to scientific rigor. They have addressed all of my previous concerns appropriately.

Reviewer #2 (Remarks to the Author):

My points have been addressed sufficiently or rebutted elegantly by the authors. Together with extensive revisions to address the points by the three reviewers, this manuscript now convincing and interesting for readers in cancer biology and all fields in biology that study non-apoptotic cell death. It convey interesting observations with respect to complementarity of CREBBP and EP300. I have no further requests.

Reviewer #3 (Remarks to the Author):

The authors have made a concerted effort to answer the reviewers comments and have performed additional informative experiments that have improved the study. They have also revised the text to reflect the comments and new data. With a few minor changes (see below), I believe the manuscript is now ready for publication in Nature Communications. Congratulations to the authors on a robust and impactful study!

Minor corrections:

In the text (page 6) when discussing Liproxstatin it says that “Here, it and almost completely rescued viability in 697KI cells exposed to low-dose Venetoclax”, but in reality, the viability goes from approx. 20 to 40%. The authors should adapt this sentence to say that the rescue is statistically significant but not a complete rescue.

We have amended this sentence as follows:

Conversely, exposure to Liproxstatin 1, which specifically inhibits ferroptosis-associated lipid peroxidation, showed significant rescue only in 697^{KI} cells

In the text (page 7) it says “697KI cells had markedly higher levels of pro-ferroptotic polyunsaturated fatty acids (PUFAs)”, this text should be altered because PUFAs are not pro-ferroptotic, they are more vulnerable to ROS damage and lipid peroxidation, which leads to ferroptosis, so the text should reflect this.

We have amended this sentence as follows:

697^{Kl} cells had markedly higher levels of polyunsaturated fatty acids (PUFAs) in their structural lipid species, which are more vulnerable to ROS damage and lipid peroxidation

Figure 4i is independent data from the other graphs in the row and needs a key - it doesn't currently have one.

Thank you for highlighting the missing key. This figure has been adjusted.